

# Improving Collisional Growth in Lagrangian Cloud Models: Development and Verification of a New Splitting Algorithm

Johannes Schwenkel[1], Fabian Hoffmann[1,*], and Siegfried Raasch[1]

[1]Institute of Meteorology and Climatology, Leibniz Universität Hannover, Hannover, Germany
[*]Current affiliation: Cooperative Institute for Research in Environmental Sciences (CIRES), University of Colorado, Boulder, Colorado, USA; NOAA Earth System Research Laboratory (ESRL) Chemical Sciences Division, Boulder, Colorado, USA

**Correspondence:** Johannes Schwenkel (schwenkel@muk.uni-hannover.de)

**Abstract.** Lagrangian cloud models (LCMs) are used increasingly in the cloud physics community. They not only enable a very detailed representation of cloud microphysics but also lack numerical errors typical for most other models. However, insufficient statistics, caused by an inadequate number of Lagrangian particles to represent cloud microphysical processes, can limit the applicability and validity of this approach. This study presents the first use of a splitting and merging algorithm designed to improve the warm cloud precipitation process by deliberately increasing or decreasing the number of Lagrangian particles under appropriate conditions. This new approach and the details of how splitting is executed are evaluated in box and single-cloud simulations, as well as a shallow cumulus test case. The results indicate that splitting is essential for a proper representation of the precipitation process. Moreover, the details of the splitting method (i.e., identifying the appropriate conditions) become insignificant for larger model domains as long as a sufficiently large number of Lagrangian particles is produced by the algorithm. The accompanying merging algorithm is essential to constrict the number of Lagrangian particles in order to maintain the computational performance of the model. Overall, splitting and merging did not affect the life cycle and domain-averaged macroscopic properties of the simulated clouds. This new approach is a useful addition to all LCMs since it is able to significantly increase the number of Lagrangian particles in appropriate regions of the clouds, while maintaining a computationally feasible total number of Lagrangian particles in the entire model domain.

## 1 Introduction

Lagrangian cloud models (LCMs) are a recently developed approach to simulate cloud physics (Andrejczuk et al., 2010; Shima et al., 2009; Sölch and Kärcher, 2010; Riechelmann et al., 2012; Naumann and Seifert, 2015; Grabowski et al., 2018). These models represent microphysics by individually simulated particles, so-called superdroplets, each representing a certain number of identical real droplets. This number is called multiplicity or weighting factor. These models have been used successfully to investigate various aspects of aerosol-cloud interactions (e.g., Andrejczuk et al., 2010; Hoffmann et al., 2015; Hoffmann, 2017) or precipitation processes (e.g., Naumann and Seifert, 2016; Hoffmann et al., 2017; Dziekan and Pawlowska, 2017).

Unterstrasser et al. (2017) have reviewed all three currently available LCM collection algorithms, from which the so-called all-or-nothing algorithm (Shima et al., 2009; Sölch and Kärcher, 2010) exhibits the generally best performance. Under certain circumstances, however, e.g., an unfortunate initialization of superdroplets with equal weighting factors, even this approach





struggles to represent the precipitation process correctly. The reason for that is easily explained. Cloud droplets cover a wide range of radii from micrometers to centimeters (e.g., Rogers and Yau, 1989). This system cannot be simplified to a couple of superdroplets (with accordingly large weighting factors) to facilitate computability. In fact, a large number of superdroplets (with accordingly small weighting factors) is needed to represent this range adequately. Since this is usually not the case, a

few of the largest superdroplets may contain unrealistically the majority of all liquid water. In order to improve the statistics of these particles, Unterstrasser et al. (2017) suggested that the splitting of these particles can help to improve the representation of the precipitation process as it is already done for other microphysical processes (e.g., nucleation in ice clouds; Unterstrasser and Sölch, 2014).

The present study introduces and verifies a splitting algorithm designed to improve the precipitation process. Additionally,

an accompanying merging algorithm is proposed that is able to unite superdroplets that are not required for an adequate representation of the precipitation process. Thus, the merging algorithm is essential to improve the computational performance of the LCM. Both algorithms are tested in zero-dimensional box-simulations, a three-dimensional simulation of a single cumulus cloud, and an established shallow cumulus test case. This paper is structured as follows. The next section briefly summarizes the used collection algorithm and the basic framework of the applied LCM. Section 3 introduces the splitting and merging

algorithms, and section 4 shows results in which splitting and merging are applied. Finally, section 5 concludes the paper.

## 2 Basic Equations of the LCM

This sections gives a short overview on the LCM's basic equations. The applied LCM was initially developed by Riechelmann et al. (2012) and its current version is documented in Hoffmann et al. (2017). Besides collection, the LCM calculates diffusional growth as well as the transport of the superdroplets. These processes are coupled to the large-eddy simulation (LES) model

PALM (Maronga et al., 2015), which solves the non-hydrostatic incompressible Boussinesq-approximated Navier-Stokes equations, and prognostic equations for the water mixing ratio and potential temperature. In addition to these coupled simulations, we use a zero-dimensional box model, in which collision and coalescence are considered as the only microphysical process.

In the following, the applied collection algorithm will be summarized to show how collection affects a superdroplet's weighting factors and, thus, to understand how collection and splitting interact. The reader is referred to Unterstrasser et al. (2017)

for a more rigorous description of this so-called all-or-nothing approach and to comparisons with other LCM collection algorithms. For the following, it is assumed that all superdroplets are sorted by their weighting factor such that $A_n > A_{n+1}$ (the case of $A_n = A_{n+1}$ will be discussed further below). For all superdroplet combinations with $1 \leq n < m \leq N_p$, where $N_p$ is the number of superdroplets located in a grid box, the probability that one droplet of superdroplet $m$ collects an arbitrary droplet of superdroplet $n$ is given by

$$p_{mn} = K(r_m, r_n)\frac{\Delta t}{\Delta V} \cdot A_n, \tag{1}$$

where $\Delta t$ is the length of the collection time step, $\Delta V$ the volume of the grid box, $r_n$ the radius of a droplet represented by superdroplet $n$, and $K$ is the collection kernel (based on Hall (1980) for this study). Since $p_{mn}$ is usually smaller than one,





collections only occur if $p_{mn} > \xi$, where $\xi$ is a random number uniformly chosen from the interval $[0, 1]$. This probabilistic approach ensures that the number of collections calculated in the model is identical to the number of collections resulting from Eq. (1) if averaged over a sufficiently long period of time.

If a collection takes place, each droplet of superdroplet $m$ will collect one droplet of superdroplet $n$. This results in commen-
surate changes in the weighting factor $A_n$ and the individual droplet mass $m_m = A_m \cdot 4/3 \pi \rho_l r_m^3$ with the liquid water density
$\rho_l$, while $A_m$ and $m_n$ remain unchanged:

$$\widehat{A}_m = A_m \qquad \text{and} \qquad \widehat{A}_n = A_n - A_m, \tag{2}$$

$$\widehat{m}_m = m_m + m_n \quad \text{and} \quad \widehat{m}_n = m_n, \tag{3}$$

where $\widehat{(..)}$ marks the variable after collection.

If $A_m = A_n$, the above-described collection would result in one superdroplet with a zero weighting factor. To avoid deleting this superdroplet, the droplets of the superdroplet that has grown by collection are distributed equally among the involved superdroplets $m$ and $n$:

$$\widehat{A}_m = \widehat{A}_n = A_m / 2, \tag{4}$$

$$\widehat{m}_m = \widehat{m}_n = 2\, m_m. \tag{5}$$

Diffusional growth is described by

$$r_n \frac{dr_n}{dt} = \frac{S}{F_k + F_d} f(r_n). \tag{6}$$

The ventilation effect $f(r_n)$ describes the accelerated evaporation of large drops. $S$ is the supersaturation (calculated in the LES), and $F_k$ and $F_d$ are coefficients considering the effects of heat conduction and the diffusion of water vapor, respectively (see, e.g., Rogers and Yau, 1989). Note that curvature and aerosol solute effects as well as gas-kinetic effects are neglected in
(6), but this equation is appropriate for the purpose of this study which focuses on the precipitation process, i.e., larger droplets for which these processes are irrelevant.

Transport of each superdroplet is described by

$$\frac{dX_n}{dt} = u(X_n) + \widetilde{u}_n, \tag{7}$$

where $X_n$ is the location of the superdroplet, $u$ the LES resolved velocity interpolated to the superdroplet's location, and $\widetilde{u}$ is a
stochastic velocity component to parameterize subgrid-scale fluctuations not resolved in the LES (see, e.g., Sölch and Kärcher, 2010).

## 3   Splitting and Merging

The following subsections will introduce techniques for the interactive modification of the number of superdroplets by splitting and merging. This is fundamentally different from most previous LCM approaches, in which the number of superdroplets is



set in the beginning of the simulation and remains constant thereafter (unless precipitation scavenges superdroplets). Note that the splitting and merging algorithms will be tested for the all-or-nothing collection approach, but they are similarly applicable to the average-impact approach introduced by Riechelmann et al. (2012).

## 3.1 Splitting

Splitting takes place if a superdroplet fulfills certain criteria. First, the radius of the superdroplet needs to be greater than or equal to a threshold $r_{\mathrm{spl}}$. This is necessary to limit splitting to the region of interest, i.e., coalescing droplets for which an improved statistical representation is required. Second, the weighting factor of the superdroplet needs to be greater than or equal to a threshold $A_{\mathrm{spl}}$ to avoid excessive and potentially useless splitting. And finally, $A_{\mathrm{spl}}$ is required to be at least larger than $\eta_{\mathrm{spl}}$, which is the number of superdroplets in which the superdroplet is split. This ensures that no superdroplets with an
unrealistic weighting factor of less than 1 are created.

The numerical implementation of the splitting can be understood as a cloning of the superdroplet that has been determined to be split. In addition to the already existing superdroplet, $\eta_{\mathrm{spl}} - 1$ new superdroplets are created. To conserve the total amount of represented droplets, the weighting factor of these $\eta_{\mathrm{spl}}$ superdroplets is reduced to

$$A_n^* = \frac{A_n}{\eta_{\mathrm{spl}}}. \tag{8}$$

Note that all $\eta_{\mathrm{spl}}$ superdroplets have identical properties immediately after splitting, including their location. However, each superdroplet will develop an individual trajectory independent from the others due to the stochastic velocity component in (7), which is determined individually for each superdroplet.

In a first straightforward approach, the thresholds $r_{\mathrm{spl}}$, $A_{\mathrm{spl}}$, and the splitting factor $\eta_{\mathrm{spl}}$ are explicitly prescribed. In the following this method is abbreviated with $S$-mode, where the $S$ stands for simple.

In a more advanced method (abbreviated $G$, standing for gamma distribution), the threshold $A_{\mathrm{spl}}$ and the splitting factor $\eta_{\mathrm{spl}}$ are estimated from an idealized gamma distribution, which is assumed to describe the distribution of droplets larger than $r_{\mathrm{spl}}$ in each grid box of the simulated model domain (e.g., Ulbrich, 1983):

$$n(r) = N_0 r^\mu \exp(-\lambda r), \tag{9}$$

where $n(r) \cdot \mathrm{d}r$ states the number of particles per unit volume in the size range $(r, r + \mathrm{d}r)$. Here, $N_0$ is the intercept, $\mu$ is the
shape, and $\lambda$ is the slope parameter of the gamma distribution. These parameters are calculated as

$$N_0 = \frac{N_r}{\Gamma(\mu+1)} \lambda^{\mu+1}, \tag{10}$$

$$\lambda = \left[ \frac{\pi \rho_\mathrm{l}}{6} (\mu+3)(\mu+2)(\mu+1)\overline{x_\mathrm{r}}^{-1} \right]^{\frac{1}{3}}, \tag{11}$$

and

$$\mu = \frac{(1-\zeta)n+1}{\zeta-1}, \tag{12}$$





where $N_r$ is the number concentration of droplets with $r \geq r_{\text{spl}}$, $\Gamma$ is the gamma function, $\rho_{\text{l}}$ is the density of liquid water, $\overline{x_{\text{r}}}$ is the mean geometric radius, and $\zeta$ is a factor calculated as

$$\zeta = \frac{M_0 M_2}{M_1^2},$$ (13)

where $M_k$ is the $k$-th moment of the mass density distribution (see Seifert, 2008). The calculation of these moments in the LCM framework will be described in section 4.1.1.

The assumed drop size distribution (DSD) is calculated from $n_{\text{bin}} = 100$ logarithmically spaced bins. (Larger values for $n_{\text{bin}}$ did not alter the results.) The center of bin $i$ is calculated as

$$r_{\text{bc},i} = 10^{(\log_{10}(r_{\text{min}}) + i\nu)},$$ (14)

where

$$\nu = \frac{\log_{10}(r_{\text{max}}) - \log_{10}(r_{\text{min}})}{n_{\text{bin}} - 1}.$$ (15)

The minimum and maximum radius of the discretized spectra are denoted with $r_{\text{min}}$ and $r_{\text{max}}$, respectively. Here, these values are set to $r_{\text{min}} = r_{\text{spl}}$ and $r_{\text{max}} = 5\,\text{mm}$, which ensures that the whole spectrum is included. Furthermore, the boundaries of bin $i$ are given by $r_{\text{bb},i} = 10^{\log_{10}(r_{\text{min}}) + (i-0.5)\cdot\nu}$ and $r_{\text{bb},i+1}$. Hence, the width of bin $i$ is $\Delta r_i = r_{\text{bb},i+1} - r_{\text{bb},i}$.

It is assumed that the weighting factor of a superdroplet should be smaller than or equal to the approximated number of droplets in the corresponding bin of the discretized gamma distribution. Thus, the weighting factor threshold is determined by

$$A_{\text{spl},i} = \max\left[n_i(r_{\text{bc},i}) \cdot \Delta r_i \cdot \Delta V,\ 1\right].$$ (16)

Accordingly, the number of newly generated superdroplets depends on the ratio of the initial weighting factor to the estimated number of droplets using the gamma distribution:

$$\eta_{\text{spl}} = \left\lfloor \frac{A_n}{A_{\text{spl},i}} \right\rfloor.$$ (17)

Since only a positive integer of superdroplets can be generated, the splitting factor is rounded down to the nearest whole number.

No matter which splitting mode is chosen, the splitting operations are executed at each time step of the LCM. Due to limited computational resources, the generation of new superdroplets must be restricted to a feasible amount. Hence, two limitations are introduced. The first restriction is the maximum splitting factor $\eta_{\text{max}}$, i.e., the maximum number of clones produced per splitting. This parameter is used for the $G$-mode, in which (17) might not be well-defined in the case of large droplets for which $A_{\text{spl},i}$ approaches zero. The second limitation ensures a computationally feasible number of superdroplets in every grid box by introducing a fixed maximum $N_{\text{P,max}}$. Accordingly, splitting operations are only executed if the number of superdroplets in one grid box is smaller than $N_{\text{P,max}}$. The latter threshold is applied for the $G$- and the $S$-mode. A suitable choice of these limits will be presented in section 4.1.2.



## 3.2 Merging

As a consequence of the potentially massive generation of new superdroplets due to splitting, the total number of superdroplets may increase sharply, which makes simulations computationally very expensive. For this reason, a merging algorithm was developed to decrease the number of superdroplets in order to reduce the required computational resources.

To avoid an impact of merging on micro- or macrophysical properties of the cloud, the algorithm is only executed in non-cloudy grid boxes (liquid water is lower than $q_l < 0.01 \, \mathrm{g \, kg^{-1}}$). Accordingly, cloudy regions, in which a high number of superdroplets is necessary for the correct representation of potential collisional growth, are left unaffected. Furthermore, it is required that the merged superdroplets are smaller or equal to $r_{\mathrm{mer}} = 0.1 \, \mu\mathrm{m}$, which ensures that only evaporated superdroplets are affected, and not raindrops that have been precipitated from the cloud. Additionally, merging is only executed in grid boxes

in which the initial superdroplet concentration is exceeded and superdroplets exhibit a weighting factor that is smaller than a certain threshold $A_{\mathrm{mer}}$, rationally chosen to be smaller or equal to the initial weighting factor. This is done to avoid decreasing the LCM's baseline capability to represent DSDs set during initialization.

    The algorithm is designed as follows. Based on the thresholds $r_{\mathrm{mer}}$ and $A_{\mathrm{mer}}$, each superdroplet with $r_m \leq r_{\mathrm{mer}}$ and $A_m \leq A_{\mathrm{mer}}$ in a non-cloudy grid box is merged with the next superdroplet of the same grid box. Here, the next superdroplet is the

superdroplet located next in the memory, which enables an efficient execution of the merging algorithm. The new weighting factor of the remained superdroplet (index $n$) is mass-weighted and given by $A_n^* = A_n + A_m \cdot r_m^3 / r_n^3$, while the other superdroplet (index $m$) is deleted. Accordingly, this leads to a new integral mass $M_n^* = M_n + M_m$, guaranteeing mass conservation. An averaging of other superdroplet properties (e.g., velocities components, radius, and location) is not implemented and probably not necessary for the correct representation of the cloud since merging is restricted to a cloud-free environment.

The use of the merging algorithm inside the certain regions of the cloud where collection plays only a subordinate role is also conceivable. However, the (probably sophisticated) determination of necessary thresholds is not within the scope of this study.

## 4 Applications

### 4.1 Box Model Simulations

In the following box simulations, the sensitivity of the LCM collection process to the number of simulated superdroplets, different splitting approaches, and their specific parameters is investigated.

#### 4.1.1 Setup

The grid box is isotropically spaced with $\Delta x = \Delta y = \Delta z = 20 \, \mathrm{m}$. The simulation time is $3600 \, \mathrm{s}$ with a constant time step of $1 \, \mathrm{s}$. To ensure adequate statistics, 25,344 boxes are calculated, and results are averaged over this ensemble. (The number

of ensemble members represents the maximum amount of grid boxes which can be calculated on 4 computing nodes in an appropriate time.) In the following this method is referred as single-box model.





Besides the traditional single-box approach a new multi-box approach is introduced. In contrast to the calculation of independent grid boxes, the multi-box approach allows superdroplets to move from one grid box to the next by prescribing a stochastic velocity (but no mean motion) in (7). The stochastic velocity component is chosen is such a way that it corresponds to a kinetic energy dissipation rate of $\epsilon_{\mathrm{box}} = 0.01\,\mathrm{m^2 s^{-3}}$, which is typical for shallow cumulus clouds (e.g., Shaw et al., 1998).

This multi-box approach has one distinct advantage over the ensemble mean of the same amount of box model simulations (single-box model), which results from the difficulties to initialize a DSD with superdroplets of a constant weighting factor, as it is done in most applications of LCMs in the literature. A single box model simulation suffers crucially from this initialization method due to a wrong representation of the largest and rarest superdroplets (Unterstrasser et al., 2017, their Fig. 17). In doing so, the rarest and largest, and therefore most important superdroplets for the collection process, are a priori over- or

underestimated. An exchange of superdroplets between the collection boxes helps to mitigate this problem. Moreover, this new approach is closer to the representation of collection in three-dimensional simulations, in which a superdroplet is not bound to a single grid box.

The impact of different numbers of superdroplets per grid box and the use of splitting for the traditional single-box approach will be discussed first; then, the impact of the new introduced multi-box approach will be presented for both splitting and

non-splitting cases. Box model simulations will be compared to the results of Wang et al. (2007), who used a high-resolution bin model. The purpose of this study is, however, not the exact reproduction of these results but a computationally efficient approximation to them using splitting. Accordingly, the initialization of the box simulation follows Wang et al. (2007), using an exponential initial DSD:

$$n(r, t = t_0) = \frac{3N_{\mathrm{init}}}{r_0^3} \cdot r^2 \exp\left(-\frac{r^3}{r_0^3}\right), \tag{18}$$

where $N_{\mathrm{init}} = 300\,\mathrm{cm^{-3}}$ is the droplet number concentration. The initial mean radius is $r_0 = 9.3\,\mathrm{\mu m}$, which leads to a liquid water content of $L_0 = 1\,\mathrm{g\,m^{-3}}$. Following Wang et al. (2007), we set the minimum droplet radius to $r_{\mathrm{min}} = 1.5\,\mathrm{\mu m}$. superdroplet radii are then selected by a random generator which follows the distribution given by (18). All superdroplets receive the same initial weighting factor:

$$A_{\mathrm{init}} = \frac{N_{\mathrm{init}} \cdot \Delta V}{N_{\mathrm{P}}}, \tag{19}$$

which ensures the number concentration of $300\,\mathrm{cm^{-3}}$. This method is also described as $\nu_{\mathrm{const}}$-init in Unterstrasser et al. (2017), which has been chosen in this study to resemble the initialization of superdroplets in less-idealized applications but also significantly hinders collisional growth.

In addition to analyzing the DSD directly, the temporal development of the zeroth and second moment of the mass density distributions is examined. Due to mass conservation in all applied approaches, the first moment is constant in time and will not

be shown. The moments of the mass distribution $f_m$ are defined as

$$M_k = \int m^k f_m(m)\,\mathrm{d}m, \tag{20}$$





where $m$ is the mass and $f_m(m)$ denotes the number concentration distribution. Note that the zeroth moment $M_0$ is the number concentration and the second moment $M_2$ is proportionate to the radar reflectivity, and thus highly sensitive to the largest droplets in the DSD.

For a given superdroplet ensemble the moments for each grid box are calculated with

$$M_k = \sum_{n=1}^{N_{\mathrm{P}}} A_n m_n^k / \Delta V, \tag{21}$$

where $m_n$ is the single droplet mass ($m_n = 4/3\pi\rho_l r_n^3$) of a superdroplet.

### 4.1.2 Box Model Results

First, the sensitivity of the collision algorithm to the number of superdroplets is examined using the LCM as single-box model. Second, the improvement of a splitting method on the collisional growth for this approach is evaluated. Subsequently, those

investigations are repeated for the multi-box approach, where superdroplets are not fixed to a certain grid box, but instead experience a stochastic motion between the grid boxes.

**Single-Box Approach**

Figures 1 and 2 show the mass density distribution after $3600\,\mathrm{s}$ and the temporal development of the moments for the LCM applied as a single-box model. The results are averaged over the entire ensemble of simulated realizations. Each grid box is

initialized with a different number of superdroplets (colored lines). The reference solution of Wang et al. (2007) is shown as a black solid line.

In Fig. 1, a significant deviation of the mass density distribution from the reference solution can be seen for all configurations. An excessively pronounced first maximum is found for all superdroplet concentrations, while the second maximum is too small and occurs too far to the left. Also, oscillations occur for radii larger than $100\,\mu\mathrm{m}$, resulting from insufficient superdroplet

statistics in this range. However, as the initial number of superdroplets increases, the depletion of the first maximum and the development of the second maximum is reproduced better. Figure 2a shows that in all cases the decrease in the number concentration is underestimated. Also for the second moment (Fig. 2b), values are predicted too low in nearly all cases. All in all, it can be observed that an increase in the number of superdroplets leads to a better agreement of the results with the bin-model even though difference are still significant for 1000 superdroplets per grid box.

In Fig. 3 and 4 the mass density distribution after $3600\,\mathrm{s}$ and the temporal development of the moments applying the splitting algorithm in different configurations are shown. Again, the splitting modes are abbreviated with $S$ for the simple splitting method and $G$ for using the splitting method based on a gamma distribution. The number following $S$ or $G$ indicates the splitting radius in microns. For all simulations, the maximum permissible number of superdroplets per grid box is limited to $N_{\mathrm{P,max}} = 1000$. The maximum splitting factor is $\eta_{\mathrm{max}} = 20$. By selecting these limits, which are chosen to represent the

upper limit of computationally feasible three-dimensional simulations, it is possible to obtain an estimate of the quality of the





individual splitting methods. The influence of the choice of these parameters is discussed below. All simulations are initialized with $N_P = 87$ superdroplets per grid box.

The black dashed line (*const.*) shows the reference LCM case in which no splitting is applied. Comparing the non-splitting case to splitting cases the results are significantly improved with respect to the reference solution. More precisely, the oscilla-

tions that occur for large droplet radii are successfully removed by splitting. Furthermore, a better representation of the second maximum is achieved by splitting, too. Independent of the splitting mode, simulations with the same splitting radius provide similar results. The only exception is between the simulations *G10* and *S10*, in which the the assumed gamma distribution enables effective splitting at slightly larger radii in $G10$ compared to *S10*. This results in a better agreement of *S10* with the bin reference. In general, a reduction of the splitting radius leads to an improved representation of the mass density distribution.

However, for all splitting simulations the reduction of the first maximum is underestimated, while the second maximum is only inadequately represented.

Similar conclusions are possible from Fig. 4, in which the timeseries of zeroth and second moment of the DSD are shown. The best agreement for the number concentration is achieved by *S10*, where many superdroplets are cloned at a very early stage. For all splitting configurations, the second moment shows a strong improvement in comparison to the LCM reference case

without splitting (*const.*) where this value is largely underestimated. Accordingly, splitting leads to an improved representation of the collisional growth in LCMs but there are still very large deviations from the bin reference.

These results exhibit how strongly collisional growth suffers from the initialization with a constant weighting factor, consistent with Unterstrasser et al. (2017). Since large superdroplets are initialized only in a few grid boxes, collisional growth is subject to a great variability in the different realizations among the ensemble. Due to that, the following subsection will repeat

this investigations using the multi-box approach, which reflects the collisional growth in 3D-applications more appropriately.

**Multi-Box Approach**

Figure 5 shows the mass density distribution after $3600\,\mathrm{s}$ time for different numbers of superdroplets (colored lines) using the multi-box approach without splitting. One can see that as the number of superdroplets increases, a better agreement with the bin model is achieved. Especially the simulations with 512 and 1000 superdroplets per grid box can reproduce the mass

density distribution well. However, for these cases, a stronger decrease of the first maximum is observed. This can be attributed to accelerated accretion, which is favored by the combination of a few large droplets with an overestimated weighting factor and a large number of superdroplets with radii of about $10\,\mathrm{\mu m}$. In contrast, a decelerated depletion of the first maximum and a weaker second peak are detected for simulations with a lower number of superdroplets. This results from the insufficient representation of the initial DSD, especially that of large droplets, which are crucial for effective collisional growth.

In Fig. 6, the temporal evolution of the number concentration and the second moment are shown. In simulations with a high number of superdroplets, a too strong reduction of the number concentration is predicted, and contrary the decrease of the zeroth moment is underestimated in cases with only 15 and 37 superdroplets. This tendency is also observed for the second moment. Simulations with a high number of superdroplets overestimate the reference, whereas simulations with only a few superdroplets result in too low values. However, comparing the results of the non-splitting cases (*const.*) in the single-box and



the multi-box simulations, the latter already provides improved results with respect to the bin model. The results show that this initialization artifact can be successfully mitigated by the newly introduced stochastic exchange between the grid boxes. For typical applications, however, the required amount of at least 512 superdroplets per grid box, necessary to derive satisfying results without splitting, is computationally unfeasible.

To maintain a reasonable amount of superdroplets, these box-simulations will be repeated now, using the splitting approach. Here, all parameters (initializing all simulations with 87 superdroplets per grid box) and splitting thresholds are identical as for the single-box approach described above but the superdroplets are now allowed to move between grid boxes.

Figure 7 shows the mass density distribution after $3600\,\mathrm{s}$ for different splitting configurations. Clear differences in the consistency with the bin reference solution can be seen. In particular, the simulations *S10* and *G10* show a good agreement

with the results of Wang et al. (2007). In both cases, the bimodal shape of the spectrum is represented well. However, for the other simulations, the deviation from the reference solution increases with increasing splitting radius, but less with the splitting mode. Both simulations with a splitting radius of $40\,\mu\mathrm{m}$ show no improvements in comparison to a simulation without splitting (*const.*, black dashed line), except in the right tail of the distribution. Figure 8 shows the moments for the different splitting configurations. Both plots indicate a slightly faster precipitation process than in the bin model, but the general agreement with

the reference is much higher than without splitting (Fig. 6).

All in all, it is shown that collisional growth is better represented by using the splitting method in both the single-box and multi-box simulations. Furthermore, the choice of the splitting mode is secondary, but the splitting radius is identified as the most crucial parameter. The multi-box simulations exhibit a distinct advantage over the single-box simulations. Due to the the presence or absence of sufficiently large droplets that might initiate collision and coalescence, as a result of the

initialization, collisional growth can be be overestimated in certain grid boxes while it is underestimated in others. Splitting and the subsequent stochastic exchange are able to distribute these so-called precipitation embryos among the entire ensemble where they are able to initiate collision and coalescence as sketched in Fig. 9, which would not be possible in the single-box approach.

**Sensitivity to Splitting Thresholds**

The limiting parameters of the splitting algorithm are now examined in sensitivity studies using the multi-box approach. For this purpose, the parameters of the maximum possible number of superdroplets per grid box $N_{\mathrm{P,max}}$, the maximum splitting factor $\eta_{\mathrm{max}}$, and the splitting radius $r_{\mathrm{spl}}$ are varied for the splitting mode $G$, which base state is defined as $r_{\mathrm{spl}} = 10\,\mu\mathrm{m}$, $\eta_{\mathrm{max}} = 20$, and $N_{\mathrm{P,max}} = 1000$. This base state is varied by individually changing the parameters $r_{\mathrm{spl}}$, $\eta_{\mathrm{max}}$, and $N_{\mathrm{P,max}}$.

Figure 10 a shows the mass density distributions after $3600\,\mathrm{s}$ for different values for $N_{\mathrm{P,max}}$. We find that a value of $N_{\mathrm{P,max}} =$

150 is sufficient to reach convergence for this setup. This reduction of the maximum number of superdroplets per grid box results in a reduction of the computational time by a factor of 15 compared to the simulation with $N_{\mathrm{P,max}} = 1000$.

The sensitivity studies for the maximum splitting factor show that this has no influence on the results (Fig. 10 b). An explanation for this is that the algorithm is executed at every time step and thus only the clone rate but not the absolute number of the clones is affected. More precisely, a low value of $\eta_{\mathrm{max}}$ may reduce how many clones are produced at a time step. However,





results show that this effect is negligible since a superdroplet will be cloned sufficiently fast at the subsequent time steps as long as $N_P \leq N_{P,max}$.

As shown before, the development of the spectrum is highly sensitive to the choice of the splitting radius. Figure 10c shows that the results converge with decreasing splitting radius, with no significant deviations for configurations with $r_{spl} \leq 15\,\mu m$.

This can be attributed to the fact that especially the largest droplets (in this case with radii of approximately $15\,\mu m$) are crucial for initiating the collisional growth. Accordingly, an improved representation of these droplets leads to an improved representation of the whole collisional growth process.

## 4.2  Single Cloud

### 4.2.1  Setup

In this case, we are simulating an idealized shallow cumulus cloud in form of a rising warm air bubble as in Hoffmann et al. (2017). The model domain is $1920\,m \times 7680\,m \times 3840\,m$ in x-, y- and z-direction, respectively. An isotropic grid spacing of $20\,m$ is used. The simulation time is $3000\,s$ while using a constant time step of $0.1\,s$. The warm air bubble is triggered by an Gaussian-shaped potential temperature perturbation $\theta^*$

$$\theta^*(y,z) = \theta_0 \cdot \exp\left[-\frac{1}{2} \cdot \left(\left(\frac{y-y_c}{\sigma_y}\right) + \left(\frac{z-z_c}{\sigma_z}\right)\right)\right], \tag{22}$$

where $\theta_0 = 0.4\,K$ is the maximum temperature difference, which decreases with a standard deviation of $\sigma_y = 200\,m$ and $\sigma_z = 150\,m$ in y- and z-direction, respectively. The center of the bubble is set to $y_c = 3840\,m$ and $z_c = 170\,m$. Due to the two-dimensional character of the temperature excess, the initial temperature perturbation is elongated homogeneously along the x-axis.

The initial profiles for temperature and specific humidity are based on the shallow cumulus case by vanZanten et al. (2011).

Note that no background winds, large-scale forcings, or surface fluxes are considered. The superdroplets are released at the beginning of the simulation and are uniformly distributed in the entire model domain. For all three directions in space, the average distance of the superdroplets is initially 4.5 m. This results in a superdroplet concentration of approximately 87 superdroplets per grid box and roughly $4.55 \cdot 10^8$ superdroplets in total. Using a weighting factor of $A_{init} = 9.0 \cdot 10^9$, an initial cloud condensation nuclei (CCN) concentration of $100\,cm^{-3}$ is represented. Additionally, simulations with 15 and 186 superdroplets

per grid box are carried out, in which the weighting factor is adjusted such that the CCN concentration of $100\,cm^{-3}$ is retained. If merging is applied, only superdroplets with a radius smaller than $r_{mer} = 0.1\,\mu m$ and with a weighting factor smaller than $A_{mer} = A_{init}/2$ are allowed to merge.

At the surface, superdroplets are absorbed if their radius is larger than $1.0\,\mu m$. For smaller particles, a reflection boundary condition is assumed to avoid that the surface acts as a CCN sink. Horizontal boundaries are prescribed with cyclic conditions.

Moreover, for collision and coalescence, the kernel by Hall (1980) is used. An overview of all conducted simulations is given in Table 1.





**Table 1.** Summary of the main parameters for the single cloud simulations.

| Simulation | $N_P$ | initial weighting factor | Splitting | $r_{spl}$ | $N_{P,max}$ | $\eta_{spl/max}$ | Merging |
|---|---|---|---|---|---|---|---|
| *const. $N_P$15* | 15 | $5.0 \times 10^{10}$ | no | - | - | - | no |
| *const. $N_P$87* | 87 | $9.0 \times 10^{9}$ | no | - | - | - | no |
| *const. $N_P$186* | 186 | $4.3 \times 10^{9}$ | no | - | - | - | no |
| *S10* | 87 | $9.0 \times 10^{9}$ | yes | $10\,\mu$m | 150 | 20 | no |
| *S20* | 87 | $9.0 \times 10^{9}$ | yes | $20\,\mu$m | 150 | 20 | no |
| *S20 merging* | 87 | $9.0 \times 10^{9}$ | yes | $20\,\mu$m | 150 | 20 | yes |
| *G20* | 87 | $9.0 \times 10^{9}$ | yes | $20\,\mu$m | 150 | 20 | no |
| *G20 merging* | 87 | $9.0 \times 10^{9}$ | yes | $20\,\mu$m | 150 | 20 | yes |

### 4.2.2 Single Cloud results

**Microphysical Properties**

Figure 11 shows the cloud averaged mass density distribution at $t = 1800\,$s for the configurations listed in Tab. 1. The left part of the spectrum is reproduced quantitatively consistent in all cases. This implies that both the splitting and the merging process

have no artificial impact on the diffusional growth process. However, the right tail of the DSDs differs significantly when the splitting algorithm is applied. The biggest drops are almost $350\,\mu$m smaller for the reference case (black lines) compared to simulations with splitting. Furthermore, splitting effectively reduces the fluctuations which occur in the reference cases for radii above $100\,\mu$m. The mass density distributions imply that the choice of the splitting mode does not affect cloud microphysical results. Likewise, the simulation $S10$, in which the splitting radius is reduced to $r_{spl} = 10\,\mu$m, shows almost no difference in the

mass density distribution compared to cases with $r_{spl} = 20\,\mu$m. Thus, it can be deduced that a splitting radius of $r_{spl} = 20\,\mu$m is sufficient for this cloud. Further investigations (not shown) in which $r_{spl}$ is successively increased to $30\,\mu$m show that a larger splitting radius leads to strong deviations from simulations with smaller splitting radius. This indicates that droplets with radii larger than $20\,\mu$m need to be represented in a statistically sufficient way to initiate the precipitation process correctly. It should be emphasized, however, that these results are only valid for a cloud with a relatively strong diffusional growth. A reduction of

the splitting radius might be required for settings in which collisions dominate the droplet's growth at smaller radii as it is the case in the previously presented box-simulations.

This behavior can be ascribed to different requirements on the superdroplet number for the convergence of different growth processes. The left part of the spectrum is dominated by diffusional growth which can be sufficiently represented by just a couple of superdroplets per grid box. By contrast, collisional growth is highly sensitive to the superdroplet number and the

correct representation of large droplets. An improved representation of these droplets is ensured by the splitting algorithm, no matter what splitting mode is used.



The improved statistics of large superdroplets are also shown in Fig. 12, where the absolute number of superdroplets per logarithmic radius ($\log(r)$) bin is presented. It is noticeable that in the reference simulations, this number decreases significantly for larger droplets (starting from a radius of approximately $r = 20\,\mu m$). In simulations in which no splitting operations are carried out, the largest droplets are represented by only a few tens of superdroplets in the whole model domain. For the $S$-mode, the superdroplet concentration is kept almost constant (except in the right tail) for all splitting cases. For the $G$-mode a second maximum at $100\,\mu m$ can be observed. This can be related to the calculation of the splitting criterion. The approximation of the mass density distribution by a gamma distribution results in a somewhat lower splitting factor for superdroplets close to the splitting radius in comparison to the $S$-mode, which shifts the superdroplet production to larger radii in the $G$ mode.

**Macrophysical Properties**

In Fig. 13, the development of the cloud is shown in timeseries of several macroscopic properties. The behavior of the different splitting configurations can be clearly seen in Fig. 13a, which depicts the ratio of the current superdroplet number to its initial value. In simulations without splitting, the superdroplet number remains nearly constant. A clear increase in the superdroplet number can be observed when splitting is used, with maximum increase of about $15\,\%$ for *S10*. In all other splitting cases, the increase in superdroplet number is notably lower and starts approximately $500\,s$ later, which corresponds to the larger splitting radius of $r_{\mathrm{spl}} = 20\,\mu m$. The lowest increase in superdroplet number is observed in the merging cases in which the maximum number of superdroplets is reached during the growing phase of the cloud and decreases in the dissipation stage.

Figure 13b and 13c show the temporal evolution of the liquid water path (LWP) and the rain water path (RWP). The RWP is defined as the integral mass of all droplets with $r \geq 40\,\mu m$. It is notable that the LWP is the same for all simulations, which emphasizes the mass conserving character of the splitting algorithm and its negligible impact on the general development of the cloud. All splitting configurations show higher RWPs in comparison to the reference runs without splitting. This increase of up to 12% is a direct result of the improved collisional growth process in the splitting configurations, resulting in more numerous and larger rain drops. This is also observed for the radar reflectivity (Fig. 13d), which is proportional to the second moment of the DSD and hence more sensitive to larger droplets.

Figure 13e and 13f display the precipitation rate and the total precipitation reaching the ground. The precipitation rate in the reference simulations without splitting exhibit high temporal variances (black lines). Those variances are successfully reduced in all splitting simulations. This can be explained by the better representation of precipitation in the splitting simulations by a larger number of superdroplets, resulting in a more uniform removal of liquid water by precipitation. As expected from the RWP, splitting slightly increases the total precipitation.

Figure 14 shows the effect of splitting on the spatial distribution of rain after $2100\,s$ simulated time for the $N_P 87$ simulation (left panel) and the *S20* splitting simulation (right panel). Similar to the reduced temporal variance in the time series of the precipitation rate (Fig. 13e), the spatial variance is also significantly reduced using splitting. Again, the precipitation is represented by only few superdroplets in the simulation without splitting, which leads to very high, localized precipitation rates. Due to splitting, raindrops with large weighting factors are split into several superdroplets with smaller weighting factors, resulting in the more realistic spatial representation of the precipitation.




All in all, the splitting of large droplets, which results in an improved representation of the collision process and thus the DSD, also partly influences the macroscopic properties of the cloud. In particular, rain water content, radar reflectivity and precipitation rate are represented in a more realistic manner. Due to the improved statistics, the temporal and spatial variance of these parameters is significantly reduced. However, the whole cloud life-cycle, which is driven by the general dynamics and
thermodynamics, is largely unaffected by splitting. Additionally, the merging shows no influence on the physical outcomes, but it allows a massive reduction of the number of superdroplets, reducing the computing time by 18% and the storage demand (which is proportional to the number of superdroplets) by at least by 7% compared to simulations applying only splitting (Fig.13a).

### 4.3 Cloud Field

### 4.4 Setup

The setup for simulating a shallow cumulus field is based on the LES intercomparison study by vanZanten et al. (2011), using their initial profiles for potential temperature and water vapor mixing ratio, the large-scale forcings, and surface fluxes. As in the original, the model domain covers an area of about $12.8\,\mathrm{km} \times 12.8\,\mathrm{km} \times 4.0\,\mathrm{km}$ in x-, y- and z-direction, respectively. The grid spacing is $\Delta x = \Delta y = 100\,\mathrm{m}$ in the horizontal, and $\Delta z = 40\,\mathrm{m}$ in the vertical. Moreover, the calculation of the domain-
averaged quantities follows (if possible) the descriptions given in the original case.

Three different simulations will be presented. In the cases *LCM $N_P$87* and *LCM $N_P$400*, the number of superdroplets per grid box are 87 and 400, respectively. With initial weighting factors of $A_{\mathrm{init}} = 1.89 \times 10^{12}$ and $A_{\mathrm{init}} = 7.0 \times 10^{12}$, respectively, these represent a CCN concentration of $100\,\mathrm{cm}^{-3}$ in each case. Moreover, one more simulation with splitting and merging is carried out. For this configuration, in which the general settings of *LCM $N_P$87* are adopted, the splitting mode $S$ with $r_{\mathrm{spl}} = 20\,\mu\mathrm{m}$,
$\eta_{\mathrm{spl}} = 20$, and $A_{\mathrm{spl}} = \Delta x \times \Delta y \times \Delta z \times 1\,\mathrm{m}^{-3} = 4.0 \times 10^5$ is used. $A_{\mathrm{spl}}$ is chosen to allow number concentrations as small as $1\,\mathrm{m}^{-3}$ to be represent by a single superdroplet.

Based on the previously presented results, the maximum number of particles per grid box is set to $N_{\mathrm{P,max}} = 150$. Merging is applied in non-cloudy grid boxes for superdroplets with a radius smaller than $r_{\mathrm{mer}} = 0.1\,\mu\mathrm{m}$ and a weighting factor smaller than $A_{\mathrm{mer}} = A_{\mathrm{init}}/2$.

### 4.5 Cloud Field Results

The analysis is focused on the influence of splitting on the macroscopic properties of the shallow cumulus field. Figure 15 shows timeseries of (a) the LWP, (b) RWP, (c) ratio of the current superdroplet number to its initial value, (d) cloud cover (cc), (e) precipitation rate, and (f) total precipitation. Despite the superdroplet number, all these parameters agree in a statistical sense. In the cases without splitting, the total superdroplet number decreases slightly in the course of the simulation due
to precipitation (Fig. 15c), while the simulation with splitting increases the total superdroplet number by about 15%. Note, however, that both LWP and RWP are at the top of model variability documented in vanZanten et al. (2011) (gray areas), which is in line with the results of Arabas and Shima (2013), who also used an LCM for the simulation of this shallow cumulus case.



Considering the temporal variability of the precipitation rate and total precipitation (Fig. 15e and f), no significant changes are detectible using splitting or a very high number of superdroplets in contrast to the single cloud simulations presented in the last section. This is foremost a result of the larger model domain alone, which attenuates variability simply by averaging. Nonetheless, a positive impact of splitting on the representation of precipitation can be seen in the probability density function

of the surface precipitation rate (Fig. 16). For the simulation with 400 superdroplets per grid box and the splitting simulation, the probability for very high precipitation rates is smaller by about one order of magnitude compared to the simulation *LCM* $N_P 87$. This clearly shows that extremely high precipitation rates, resulting from individual superdroplets with large weighting factors, are mitigated when splitting is applied. Accordingly, splitting is important for a statistical appropriate representation of individual rain events and necessary for the process-level understanding of the precipitation process, but the general features

of the cloud field, as it was the case for the single cloud, are largely unaffected.

## 5   Conclusions

The main objective of this paper was the development and verification of a splitting algorithm to improve collisional growth in Lagrangian cloud models (LCMs), which are known to insufficiently represented this process. This is especially the case if the number of superdroplets is low and accordingly the number of real droplets represented by each superdroplet (the so-

called weighting factor) is high, leading to a oversimplified representation of the droplet size distribution (Riechelmann et al., 2012; Unterstrasser et al., 2017). Splitting is carried out by cloning superdroplets of interest (large radius and high weighting factor) into a large number of identical superdroplets with commensurately reduced weighting factors, which improves the representation of the DSD in the desired areas. An accompanying merging algorithm has been also introduced, too. It is designed to merge two superdroplets into one, counteracting the (potentially) massive production of superdroplets due to

splitting and hence a significant increase of computational costs.

The splitting and merging algorithms have been validated using box-simulations, a simulation of a single cumulus cloud, and an established shallow cumulus test case. The box-simulations confirmed that the capability of an LCM to represent the temporal evolution of a DSD due to collision and coalescence depends crucially on the number of simulated superdroplets (Shima et al., 2009; Riechelmann et al., 2012; Unterstrasser et al., 2017). Without splitting, only simulations with more than 500

to 1000 superdroplets per grid box were acceptably reproducing literature references. By applying the new splitting algorithm, however, the results improved significantly using only up to 150 superdroplets per grid-box. Furthermore, the box-simulations revealed that the radius from which splitting is applied is the most important parameter of the splitting algorithm. A value of $15\,\mu m$, which corresponds to the typical radii of the first colliding droplets in clouds, was found to be appropriate. Other investigated parameters have shown only a minor impact on the results as long as a sufficiently large maximum number of

superdroplets is allowed to be produced by splitting ($\geq 150$).

In the idealized single cloud simulation, splitting improved the representation of collisional growth with up to 70 % larger maximum radii and a slight increase of the rain water path of up to 12 %. Moreover, splitting improves the spatial and temporal representation of precipitation by distributing the precipitable water on more superdroplets with an accordingly smaller




weighting factors. It is important to note, however, that the life cycle and domain-averaged macroscopic properties are almost not affected by the splitting process. If applied, the merging algorithm has been shown to reduce the computing time by 18% and the storage demand at least by 7% in comparison to simulations with splitting alone. Since merging is restricted to cloud-free regions, its application did not alter the simulated physics. Similar findings on the effect of splitting on the production of
5   rain have been made for the shallow cumulus test case.

In the light of the fact that LCMs become increasingly important in the field of modeling cloud microphysics, it is necessary to minimize the (typically) large demand of memory and computing time required for their application. Thus, a fixed number of superdroplets needs to be replaced by a dynamic number, which adapts interactively to the given physical and numerical requirements. In this regard, the presented methods follow the approaches by Grabowski et al. (2018), in which superdroplets
10  are only created after activation, or Naumann and Seifert (2015), who restricted the superdroplet approach to the representation of raindrops. Of course, all these approaches have their specific advantages and disadvantages, but they are necessary steps to apply LCMs in a wider range of future applications.

*Code availability.*  The LES model used in this study (revision 2263) is publicly available on https://palm.muk.uni-hannover.de/trac/browser/ palm?rev=2263. For analysis, the model has been extended and additional analysis tools have been developed. The extended code is available
15  from the authors on request.

*Acknowledgements.*  All simulations have been carried out on the Cray XC-40 systems of the North-German Supercomputing Alliance (HLRN, https://www.hlrn.de/). The publication of this article was funded by the Open Access fund of Leibniz Universität Hannover.



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





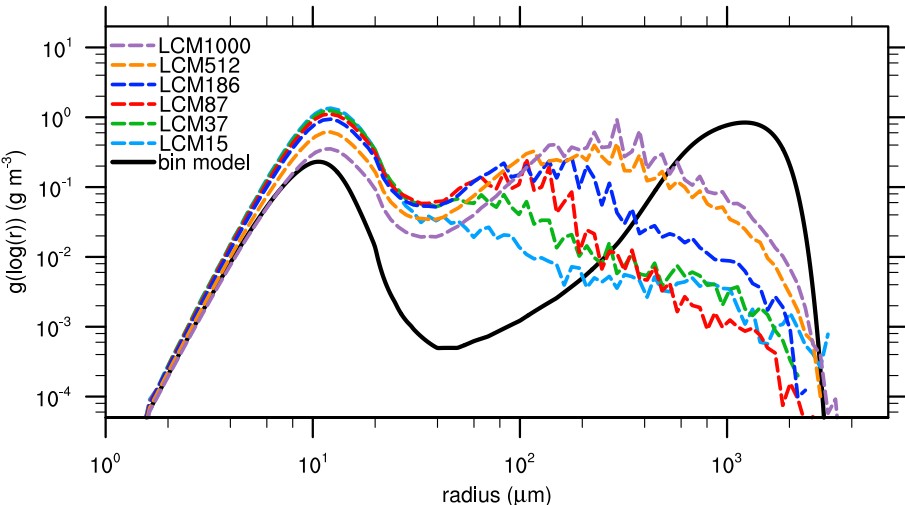

**Figure 1.** Mass density distribution for the single-box approach after 3600 s. The black solid line denotes the solution of Wang et al. (2007). The colored dashed curves show the solution of the LCM with different numbers of superdroplets per grid box.

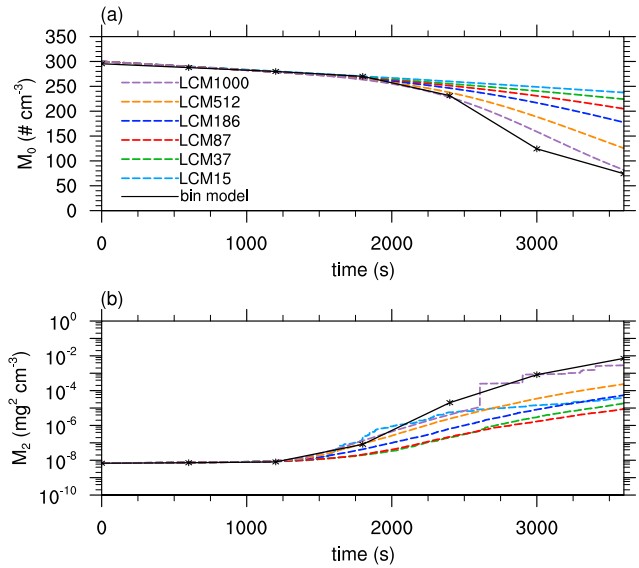

**Figure 2.** Moments of the mass density distribution as a function of time obtained from the single-box simulations. The black solid line denotes the solution of Wang et al. (2007). The colored dashed curves show the solution of the LCM with different numbers of superdroplets per grid box.



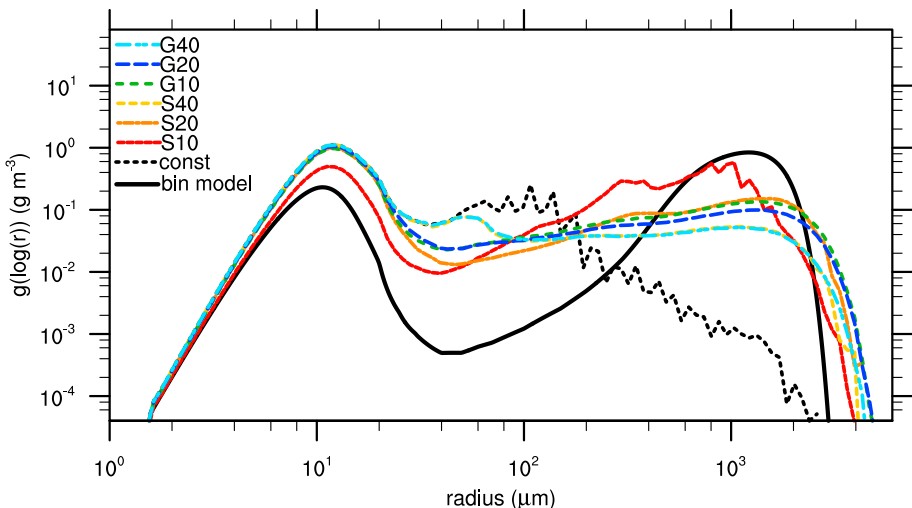

**Figure 3.** Mass density distribution for the single-box approach after 3600 s. The black solid line denotes the solution of Wang et al. (2007), the black dashed curve the reference case (without splitting). The colored dashed curves show solution for splitting simulation with different configurations.

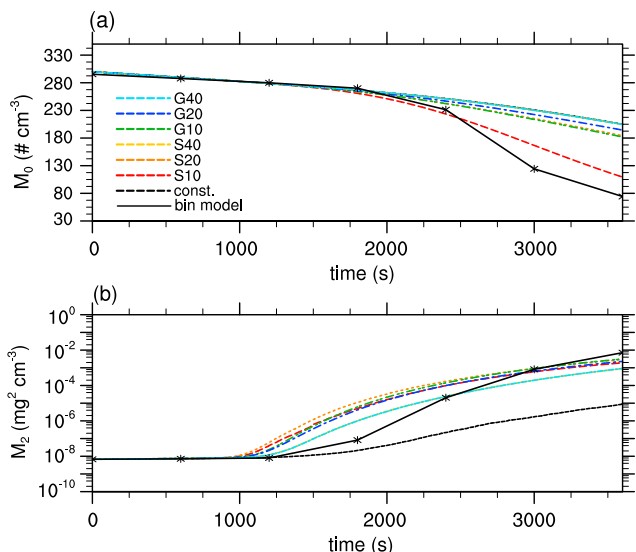

**Figure 4.** Moments of the mass density distribution as a function of time obtained from single-box simulations. The black solid line denotes the solution of Wang et al. (2007), the black dashed curve for the reference simulation (without splitting). The colored dashed curves show the solutions for different splitting configurations.



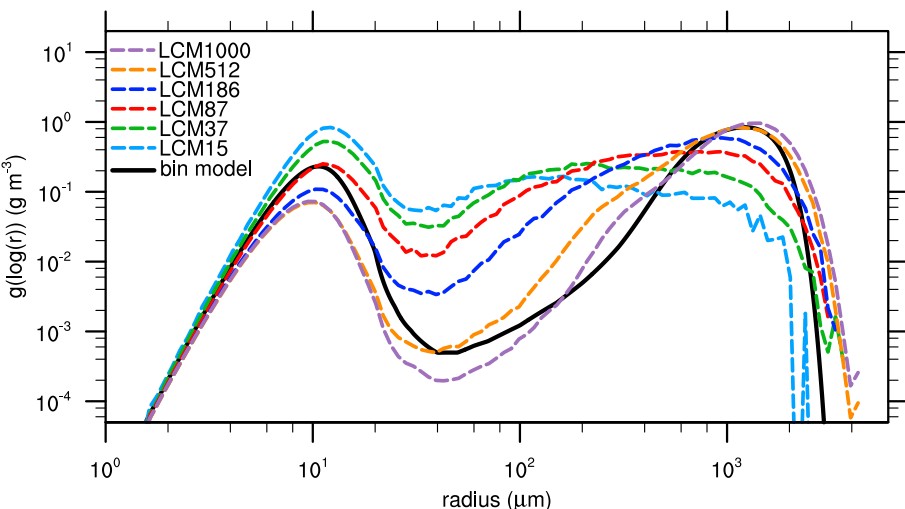

**Figure 5.** Same as Fig. 1 but for the multi-box approach, i.e. interactions between the grid boxes are possible.

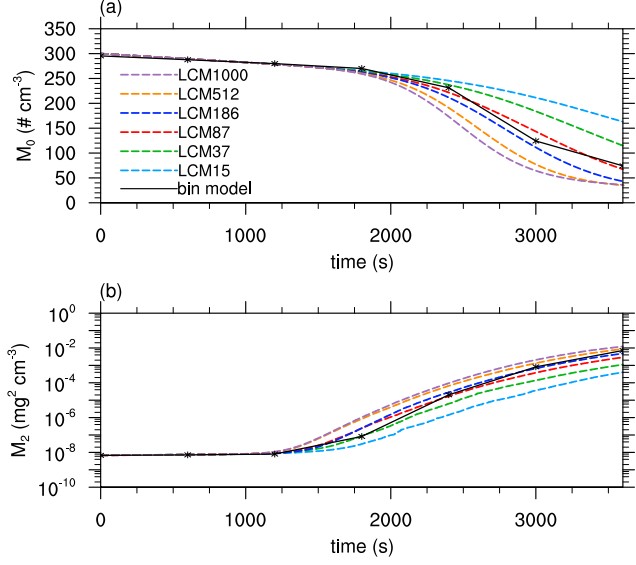

**Figure 6.** Same as Fig. 2 but for the multi-box approach, i.e. interactions between the grid boxes are possible.





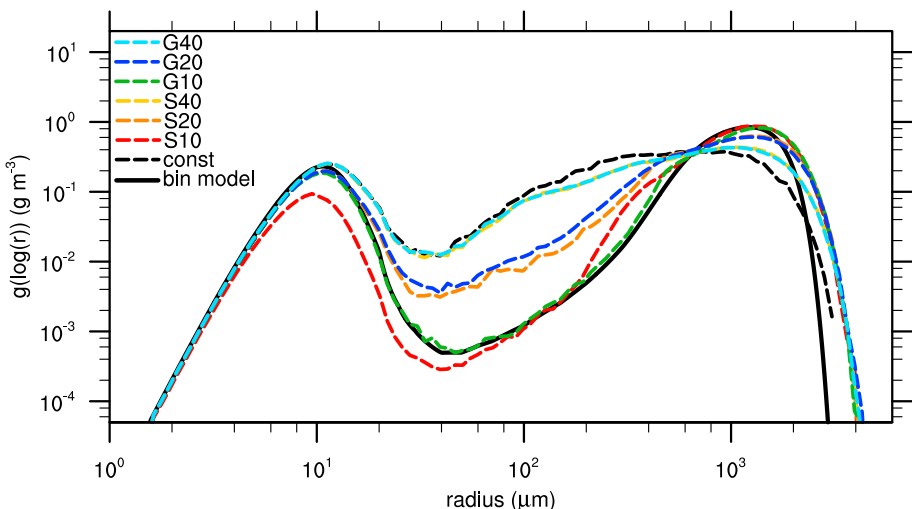

**Figure 7.** Same as Fig. 3 but for the multi-box approach, i.e. interactions between the grid boxes are possible.

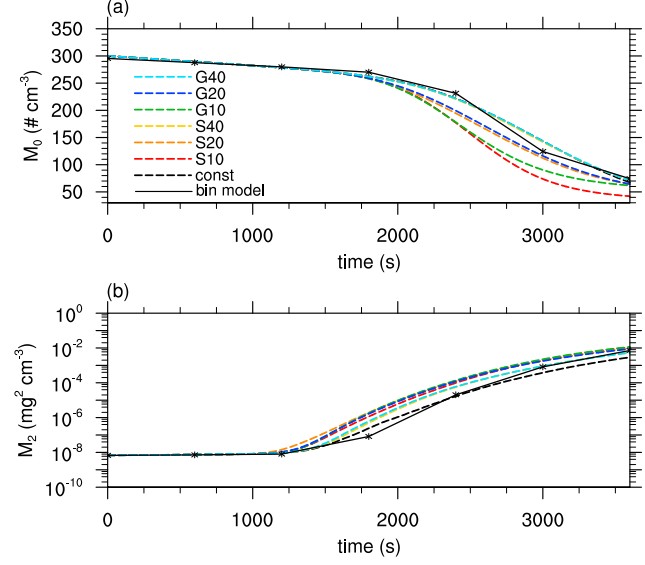

**Figure 8.** Same as Fig. 4 but for the multi-box approach, i.e. interactions between the grid boxes are possible.




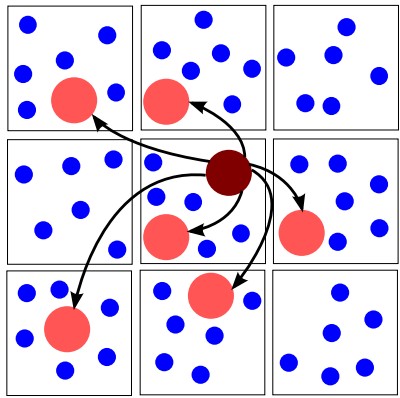

**Figure 9.** Schematic representation on how splitting affects the spatial distribution of large superdroplets. The squares outline the different grid boxes with superdroplets of the size of cloud droplets (blue) and superdroplets representing rain drops (dark red). Without splitting (left), the rain drop is represented by only one superdroplet. In the splitting case with multi-box approach, this superdroplet is cloned into several superdroplets, which are able to move in other grid boxes (due to their individual SGS-velocities) where they initiate or affect collisional growth.



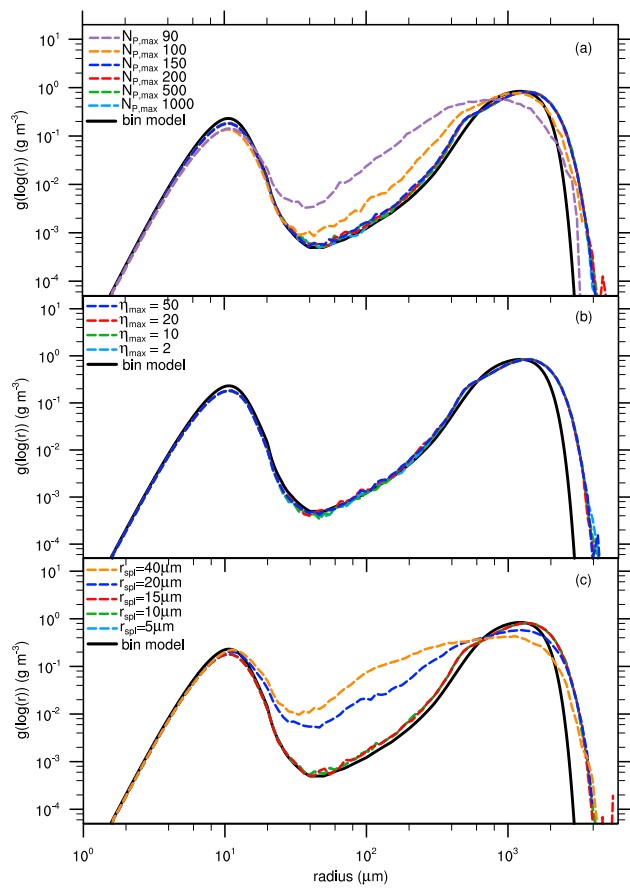

**Figure 10.** Mass density distribution for the box-simulation after 3600 s. The black solid line denotes the solution of Wang et al. (2007). In (a), sensitivity studies for different values of $N_{\mathrm{P,max}}$ are presented. In (b), simulations for different values of $\eta_{\mathrm{max}}$ are shown. In (c), results for different splitting radii are displayed. All Sensitivity studies are conducted using the splitting mode $G$.





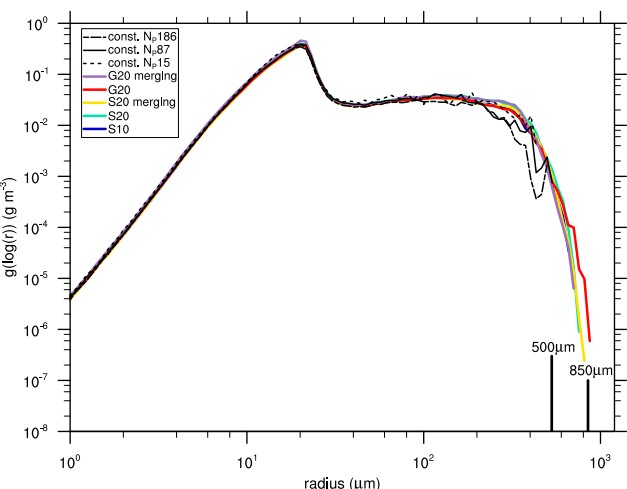

**Figure 11.** Mass density distribution after $1800\,\mathrm{s}$ for the idealized single cloud simulations using parameters described in Tab.1.

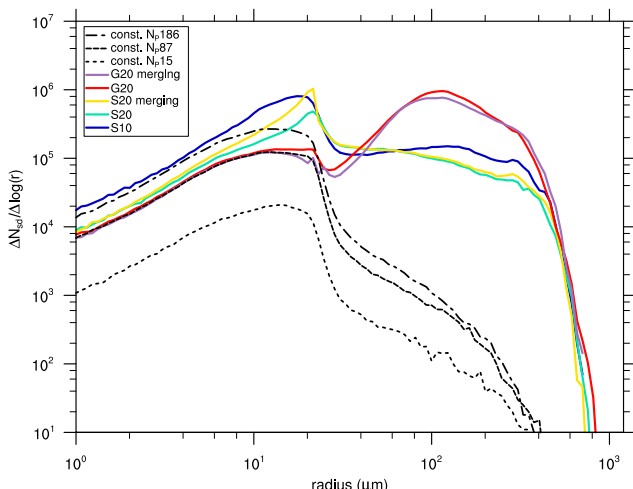

**Figure 12.** Total number of superdroplets per logarithmic radius bin after $\mathrm{t} = 1800\,\mathrm{s}$ for the idealized single cloud simulations using parameters described in Tab.1.



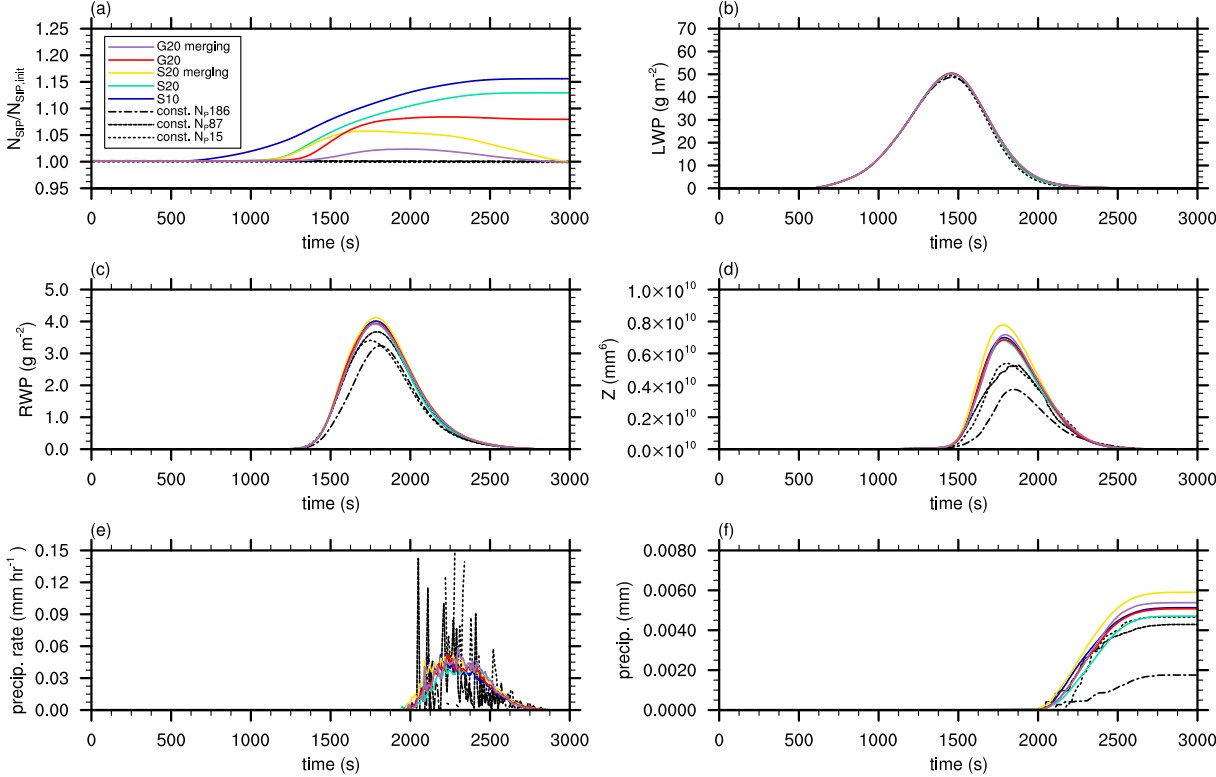

**Figure 13.** Timeseries of different variables for the idealized single cloud simulation for different initial numbers of superdroplets and splitting configurations. In (a), the ratio of the actual and initialized number of superdroplets in the whole model domain is shown. The liquid water path (LWP) and rainwater path (RWP) are displayed in panels (b) and (c), respectively. In (d), the total radar reflectivity is shown. Panels (e) and (f) show the precipitation rate and total precipitation, respectively.

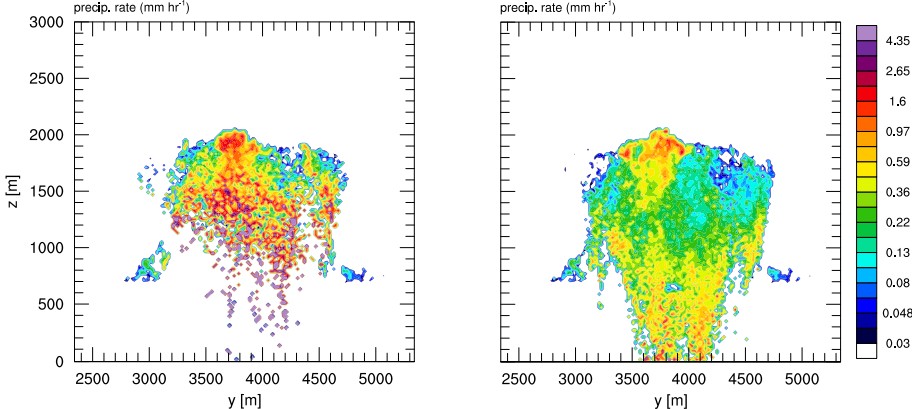

**Figure 14.** Vertical cross-sections of the precipitation rate for the reference case (left) and the splitting case *S20* (right).



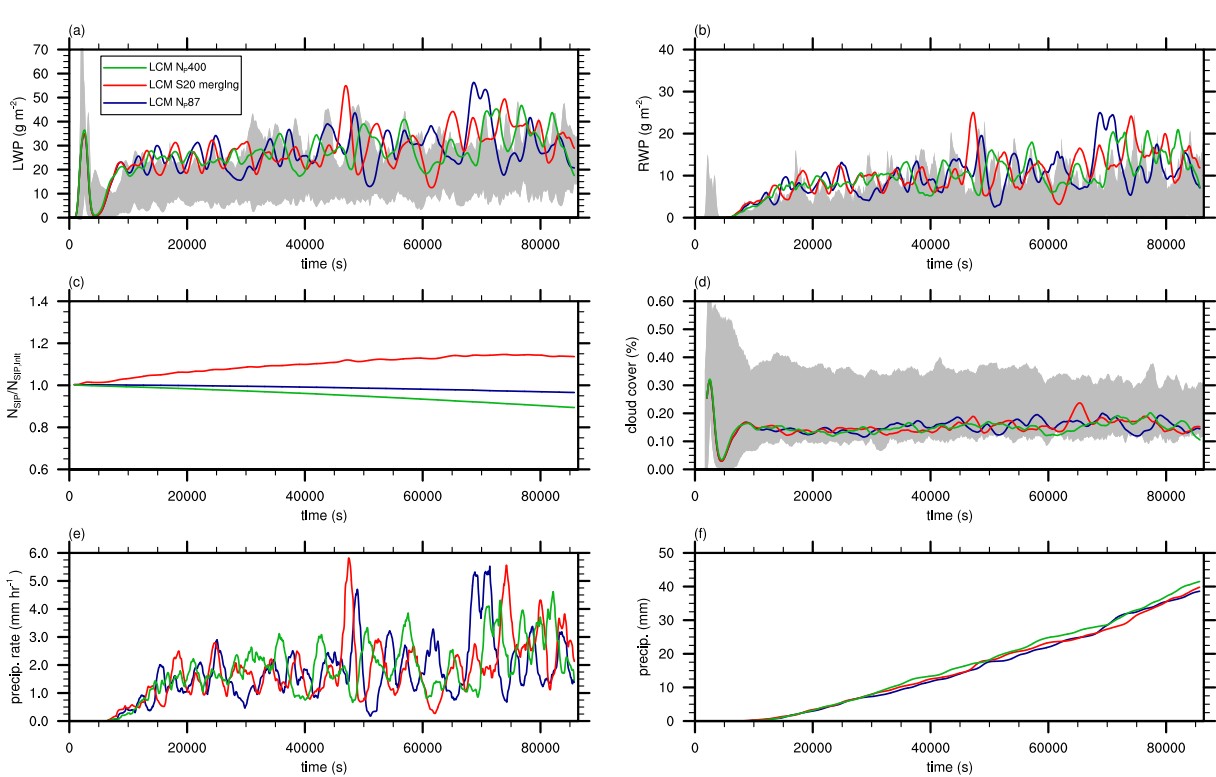

**Figure 15.** Timeseries of (a) the liquid water path (LWP), (b) rainwater path (RWP), (c) ratio of the actual number of superdroplets to the initial number of superdroplets, (d) cloud cover, (e) precipitation rate, and (f) total precipitation for different initial numbers of superdroplets and splitting configurations. The gray areas in (a), (b) and (d) indicate the documented model variability of the simulated shallow cumulus case (vanZanten et al., 2011).





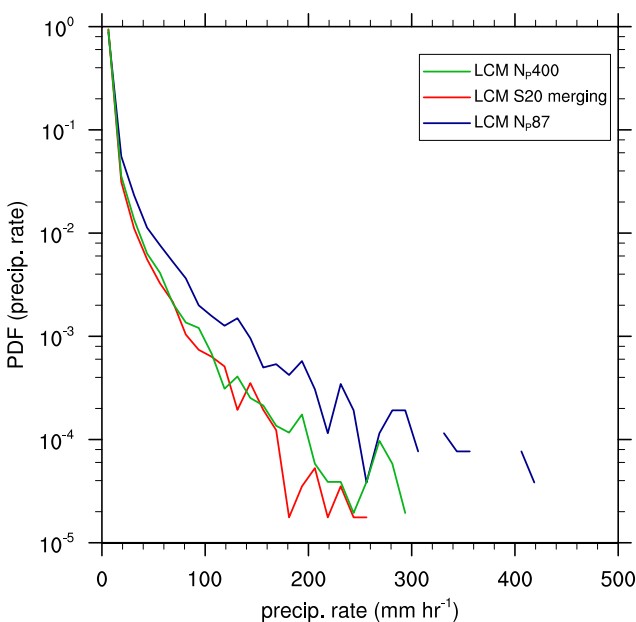

**Figure 16.** Probability density function of precipitation rates for different initial numbers of superdroplets and splitting configurations.