# Peer review of "Improving Collisional Growth in Lagrangian Cloud Models: Development and Verification of a New Splitting Algorithm"

_Geoscientific Model Development, 2018_

## Short Comment (SC1) · 11 Jun 2018

Dear authors,

in my role as Executive editor of GMD, I would like to bring to your attention our Editorial version 1.1: http://www.geosci-model-dev.net/8/3487/2015/gmd-8-3487-2015.html This highlights some requirements of papers published in GMD, which is also available on the GMD website in the 'Manuscript Types' section: http://www.geoscientific-model-development.net/submission/manuscript_types.html In particular, please note that for your paper, the following requirement has not been met in the Discussions paper:

[Figure]

- "The main paper must give the model name and version number (or other unique identifier) in the title."

Please provide the name and a version number of the new splitting algorithm and/or the LBM used in the title of your revised manuscript. Note, that a name and a version number are important to identify your specific developments.

As explained in
https://www.geoscientific-model-development.net/about/manuscript_types.html. GMD is encouraging authors to upload the program code of models (including relevant data sets) as supplement or make the code and data of the exact model version described in the paper accessible through a DOI (digital object identifier). In case your institution does not provide the possibility to make electronic data accessible through a DOI you may consider other providers (eg. zenodo.org of CERN) to create a DOI. Please note that in the code availability section you can still point the reader to how to obtain the newest version. If for some reason the code and/or data cannot be made available in this form (e.g. only via e-mail contact) the "Code Availability" section need to clearly state the reasons for why access is restricted (e.g. licensing reasons).
Especially, please note, that it is not enough, that the code will be available in the future. It must be available now and the exact version of the code published in this article needs to be made available.

Yours, Astrid Kerkweg

---

## Referee Comment (RC1) · Anonymous Referee #1 · 22 Jun 2018

Lagrangian cloud microphysics models use relatively few computational droplets (also known as super-droplets, SDs) to represent huge number of real droplets that clouds are made of. This simplification makes it difficult to model coalescence of droplets and artificialy amplifies fluctuations associated with transport and coalescence. The paper presents a novel method of mitigating these issues and is of potential interest for the GMD readers. The method proposed is to split computational droplets that represent large fraction of liquid water into couple computational droplets, each representing a smaller amount of liquid water. This new aproach is shown to improve results of simulations, especially in the idealized box models. However, some non-trivial results are only vaguely discussed, or their analysis is arbitrary. Therefore I suggest including a

more detailed discussion of the results, that would address the following points:

1. Single cloud simulations of cumulus show, that splitting increases the amount of rain water. This increase is not seen in the cloud field simulation. Authors conclude that there is no increase in rain water because of averaging over a large cloud field (p.15 l.4). I do not understand how this is relevant. If amount of rain water in each cloud is increased, the average should also be increased.

2. Judging from single cloud simulations, merging increases radar reflectivity and amount of precipitation (see Fig. 13, especially for the S20 case). I suppose that this is not the case, but that the difference comes from different model realisations. This could be clarified if a small ensemble of single cloud simulations was ran for each case discussed in the section 4.2.

3. In single cloud simulations without splitting, amount of precipitation decreases as the number of computational droplets is increased (Fig. 13). Simulations with splitting are in better agreement with $N_p = 15$ than with $N_p = 186$. Why is it so?

4. In box model simulations without splitting, largest droplets produced in LCM are larger than the Smoluchowski equation predicts, especially if number of computational particles is large (Fig. 5). On page 9, line 26, Authors argue that this is because there are few large SDs with high weighting factors. If this is the case, then shouldn't the effect decrease with increasing number of SDs? The opposite is observed - this effect is more pronounced as more SDs are added. Moreover, if the Authors' argument was correct, splitting of SDs should also fix this problem. On the contrary, LCM with splitting also produces too large droplets, as seen in Figs. 7 and 10.

Comments to the "Conclusions" section:

5. In the first paragraph of "Conclusions", Authors claim that LCMs are "known to insufficiently represent" coalescence. I suppose that they are refering to results of box model simulations, shown in Fig. 1. Not all LCM insufficiently represent coalescence, but only those that initialize SDs in the same way it is done in the paper. Other initialization procedures give much better box model results, e.g. Dziekan and Pawlowska 2017.

6. In "Conclusions", cloud field simulations are claimed to show similar effects of splitting on the production of rain as single cloud simulations. This is not true - splitting increases the amount of rain in single cloud simulations, but does not affect the amount of rain in cloud field simulations.

7. A short discussion of potential impact of SD merging on aerosol processing would be desirable.

8. Authors write that merging of SDs reduces computing time by $18\%$. How does splitting affect performance?

Technical comments:

9. In lower panels of Fig. 15, precipitation rate from vanZanten et al. 2011 should be shown.

10. Throught the paper, large variability in results is alternatively called "oscillations" or "fluctuations" (e.g. p. 8, l. 19). Oscillation is a periodic process, so I suggest the word "fluctuations" to be used.

11. In the last line on page 3, Authors write that splitting makes their approach "fundamentally different". To me this seems to be a too strong statement - "different" would suffice.

12. Why is the blue line in Fig. 16 discontinuous?

13. In Fig. 5 spectra for 512 and 1000 SDs have a local maximum on the large end. What is the cause?

---

## Referee Comment (RC2) · Anonymous Referee #2 · 3 Jul 2018

**1  Introduction**

The authors introduce a new extension of the lagrangian microphysics schemes - the splitting and merging algorithm. The splitting part of the algorithm increases the resolution of the lagrangian microphysics scheme for big droplets. This is especially important for correctly representing the collisions between droplets and the resulting onset of precipitation. The merging part of the algorithm decreases slightly the computational cost of the splitting algorithm. Both developments are described and tested. The paper is well written and interesting for the GMD community. It should be published after some corrections and additional tests.

**2 Major comments**

- A lagrangian microphysics scheme can represent collisions in different ways and can be initialized in different ways. Both choices have big impact on the accuracy of the scheme, Unterstrasser et al. (2017). Here the authors in their implementation of the lagrangian microphysics scheme use the most accurate way to represent collisions (the all-or-nothing algorithm Shima et al. (2009)). This is great. However, they initialize the droplet size distribution with constant weighting factors. This is the worst initialization strategy for representing collisions, see Figure 1 in Unterstrasser et al. (2017). It cannot be said that this is the standard way to initialize lagrangian microphysics schemes. Other groups initialize their schemes in different ways, see for example Unterstrasser and Sölch (2014), Arabas et al. (2015). Because this work focuses on improving the representation of collisions between droplets, such a bad-for-collisions initialization choice is not justifiable.

  The authors should also test their splitting and merging algorithm with a better initialization way (any of the singleSIP, multiSIP or $\nu_{random}$ from Unterstrasser et al. (2017) would suffice). Redoing the single cloud and cloud field simulations might be to expensive and unnecessary for the purpose of testing. However, it would be very interesting to see the single and multi-box tests done again with a different initialization choice. Does the splitting and merging algorithm improve the results as much for a different initialization choice? Are the multi-box simulations really necessary if the droplet size distribution is initialized correctly? How does the estimate given in line 25 on page 15 change with a better initialization?

- page 6 line 3: What is more important in counterbalancing the increase in super-droplet number due splitting algorithm? - Is it the limiting number of super-droplet per grid-box $N_{P,max}$ or merging algorithm? What is the impact of merging on the possible future activation of merged super-droplets to cloud droplets? What is the resulting resolution of the scheme for aerosol particles after merging? Is

merging more accurate than using super-droplets just for representing clouds and precipitation and parameterizing activation process, as it is done in Grabowski et al. (2018)?

- page 6 line 16-19: The mass conservation is a necessary constraint. But it is not enough to determine the super-droplet properties after merging. For the purpose of this study it's probably not important to have a more detailed strategy for determining super-droplet properties after merging. However it might be important for aerosol processing or secondary activation of aerosol particles to cloud droplets. Having that in mind, what would be the best way to determine the super-droplet properties after merging? For example: The current one assumes that the super-droplet radius after merging is equal to the radius of the super-droplet with the bigger weighting factor. An alternative could be to assume that the weighting factors are summed and to calculate the new radius from the mass conservation?

- page 7 line 11: Are all the boxes in the multi-box simulation homogeneous and there is no droplet sedimentation? If yes, what is the difference between the introduced here multi-box approach and a single-box simulation that would use a better way to initialize the droplet size distribution (with a better initial representation of the tail of the distribution) and use the same total number of super-droplets as the multi-box simulation? What is the total volume simulated in the multi-box approach vs the single-box? Is multi-box approach increasing the simulated volume only to introduce different realizations of the initial condition to counter the problems introduced by the constant weighting factor initialization?

- Both the high super-droplet concentration simulations (Fig 5) and the splitting simulations (Fig 7) overestimate the biggest droplet sizes. Why? Would using even more super-droplets allow to reach better agreement with the bin reference simulation for the biggest droplet sizes?

- page 11 line 14: What is the benefit of using a 3-dimensional simulation setup if the initial condition is 2-dimensional? Using a 2-dimensional setup (with cyclic boundary condition in the missing dimension) would reduce the computational cost by two orders of magnitude. This in turn would allow to test the performance, accuracy and convergence of the scheme for two orders of magnitude higher super-droplet concentrations.

- Figure 11: I agree that eliminating the fluctuations in the size distribution of big droplets is a good result. However, the box model tests (Fig 5 and 7) show that the lagrangian scheme overestimates the sizes in the large tail of he droplet distribution. Therefore the fact that the biggest drop size for the simulation with splitting is 350$\mu$m bigger might not necessarily be an improvement?

- Figure 11 and 12: Is there any change in the behavior for the small droplet sizes at t=3000s (end of the simulation) that is caused by merging?

- Figure 13 c, d, f: Why is the simulation with the biggest initial concentration of super-droplets (N=186) the biggest outlier? Out of the simulations without split-ting and merging I would expect the one with initial N=15 to perform the worst and not the best. If the resolution for big droplets is important for collisions then the N=186 simulation should be better than N=15? Would running an ensemble average help? Or is this behavior consistent for even higher initial super-droplet concentrations?

- page 14 line 6: What is the computational cost and storage demand increase due to splitting algorithm?

- page 15 line 13: The work by Dziekan and Pawlowska (2017) shows that when used in high-enough resolution the lagrangian methods truly do resolve the col-lisions between droplets. The tests presented in Unterstrasser et al. (2017) also

suggest that when initialized correctly and when using a good algorithm for representing collisions the lagrangian microphysics schemes can represent collisions for coarser resolution settings (i.e low initial super-droplet concentration). The splitting and merging algorithm presented here is a valid improvement. It is especially important for large eddy simulation applications when by necessity the lagrangian schemes have to be used with low super-droplet concentrations. Nevertheless in my opinion, saying that in general the lagrangian methods are known to insufficiently represent collisions is not justified.

**3   Minor comments**

- The Lagrangian particles used to represent droplets are named differently by different modeling groups: super-droplets, simulation particles, etc. The original term super-droplet was introduced by Shima et al. (2009). Instead of using another notation $superdroplet$ it would be better to follow the notation that is already used by others.

- page 1 line 16: Are the references meant to be chronologically or alphabetically ordered?

- page 1 line 16: A couple more references to lagrangian microphysics applications: Lee et al. (2014), Arabas et al. (2015), Sardina et al. (2018)

- page 1 line 23: Which of the previously cited works use all-or-nothing algorithm?

- page 2 line 3: What is a large weighting factor and a large number of super droplets? Could you provide an order of magnitude estimate of those for a typical large eddy simulation grid box?

[Figure]

- page 2 line 30: Is it more correct to say that it is a probability that super droplet *m* will collect super droplet *n*?

- page 2 line 30: An alternative is to consider for collisions only the non-overlapping pairs and scale the probability - see section 5.1.3 in Shima et al. (2009) or section 5.1.4 in Arabas et al. (2015). This allows for the collision algorithm to scale linearly and not quadratically with the number of super-droplets. Maybe it should be mentioned?

- page 6 line 14: Are the super-droplets sorted with regard to their size in memory? Or in other words are the super-droplets merged with the most similar super-droplet in a given grid-box?

- page 6 line 26: Is diffusional growth allowed in the box model simulations? If yes, what is the assumed saturation? It is confusing with regard to line 15 on page 12. - Saying that collisions dominate the droplet growth in the box simulations suggests that there are other processes considered.

- page 6 line 30 (and onward): It's a bit confusing to talk about grid boxes when using a single box model setup. There is no real computational grid here.

- page 6 line 30: Where does the assumption that the box dimensions are $\Delta x = \Delta y = \Delta z = 20$m matter for the lagrangian scheme? Is it enough to say that the box volume is $8 * 10^3$m$^3$?

- page 7 line 1: How many boxes are used? How are the boxes in multi-box approach located with regard to each other? What are the boundary conditions?

- page 7 line 6: As stated in my first major comment - I don't agree that the initialization with constant weighting factors is a standard. Because it is such a bad initialization for representing collisions the new splitting algorithm should also be tested with a better initialization.

- Figure 2b: Why is the LCM1000 behaves like a step function after 2500s?

- Figure 3: The plotting colors and patterns should be kept the same between Figs. 3,and 7 to allow easier comparison.

- page 10 line 25: Is the initial super-droplet concentration again 87 per box?

- Figure 13a and 15c: The notation $N_{SIP}$ was never used before. The authors choose to refer to the lagrangian particles as super-droplets and not simulation particles.

- page 14 line 15: When it is not possible? What happens then?

- page 15 line 24: Dziekan and Pawlowska (2017) would be a valid reference here.

- page 15 line 27: Figure 10 suggests that the maximum number of super-droplets $N_{P,max}$ is as important as the splitting radius $r_{spl}$?

- page 16 line 13: The code of the Large Eddy Simulation model along with the new splitting and merging algorithm is available online and therefore fulfills the GMD requirements. It would have been great if the simple box model tests were available as a stand alone and easy to download and compile project. It would enable easy testing of the algorithm by others, for example this reviewer. It is by far too much coding to ask to do this now. I would just like to leave this comment as an idea for future development and testing.

**References**

Arabas, S., Jaruga, A., Pawlowska, H., and Grabowski, W. W.: libcloudph++ 1.0: a single-moment bulk, double-moment bulk, and particle-based warm-rain microphysics library in C++, Geosci. Model Dev., 8, 1677–1707, https://doi.org/10.5194/gmd-8-1677-2015, 2015.

[Figure]

Dziekan, P. and Pawlowska, H.: Stochastic coalescence in Lagrangian cloud microphysics, Atmos. Chem. Phys., 17, 13 509–13 520, https://doi.org/10.5194/acp-17-13509-2017, 2017.

Grabowski, W. W., Dziekan, P., and Pawlowska, H.: Lagrangian condensation microphysics with Twomey CCN activation, Geosci. Model Dev., 11, 103–120, https://doi.org/10.5194/gmd-11-103-2018, 2018.

Lee, J., Noh, Y., Raasch, S., Riechelmann, T., and Wang, L.-P.: Investigation of droplet dynamics in a convective cloud using a Lagrangian cloud model, Meteorol and Atmos Phys, 124, 1–21, https://doi.org/10.1007/s00703-014-0311-y, 2014.

Sardina, G., Poulain, S., Brandt, L., and Caballero, R.: Broadening of Cloud Droplet Size Spectra by Stochastic Condensation: Effects of Mean Updraft Velocity and CCN Activation, J. Atmos. Sci., 75, 451–467, https://doi.org/10.1175/JAS-D-17-0241.1, 2018.

Shima, S., Kusano, K., Kawano, A., Sugiyama, T., and Kawahara, S.: The super-droplet method for the numerical simulation of clouds and precipitation: a particle-based and probabilistic microphysics model coupled with a non-hydrostatic model, Quart. J. Roy. Meteor. Soc., 135, 1307–1320, https://doi.org/10.1002/qj.441, 2009.

Unterstrasser, S. and Sölch, I.: Optimisation of the simulation particle number in a Lagrangian ice microphysical model, Geosci. Model Dev., 7, 695–709, https://doi.org/10.5194/gmd-7-695-2014, 2014.

Unterstrasser, S., Hoffmann, F., and Lerch, M.: Collection/aggregation algorithms in Lagrangian cloud microphysical models: rigorous evaluation in box model simulations, Geosci Model Dev., 10, 1521–1548, https://doi.org/10.5194/gmd-10-1521-2017, 2017.

---

## Author Comment (AC1) · 10 Aug 2018

We welcome the Executive Editor's reference to the GMD requirements. The Editor points out that the manuscript "must give the model name and version number (or other unique identifier) in the title". We understand that it makes sense to add a version number to the title in many cases. However, we think this is not meaningful in our particular case since our manuscript does not document a specific model, but a new idea: We write about the development of a new modeling methodology, its verification, and application. And this approach can be easily transferred to other models. In that spirit, our paper follows other GMD articles, which do not state a specific model name or version

number in the title, but write more generally about the development or verification of new modeling approaches (e.g., Unterstrasser et al. 2017, doi: 10.5194/gmd-10-1521-2017; Grabowski et al. 2018, doi: 10.5194/gmd-11-103-2018). On the other hand, we already state the name and version number of the applied model in the manuscript. In the "Code Availability" section, a link allows the reader to view, download, and install the developed code. All in all, our idea and the results are independent of a certain model, which is why we like to keep the title of our manuscript, without mentioning a specific model name or version number.

---

## Author Comment (AC2) · 10 Aug 2018

**Lagrangian cloud microphysics models use relatively few computational droplets (also known as super-droplets, SDs) to represent huge number of real droplets that clouds are made of. This simplification makes it difficult to model coalescence of droplets and artificially amplifies fluctuations associated with transport and coalescence. The paper presents a novel method of mitigating these issues and is of potential interest for the GMD readers. The method proposed is to split computational droplets that represent large fraction of liquid water into couple computational droplets, each representing a smaller amount of liquid water. This new approach is shown to improve results of simulations, especially in the idealized box models. However, some non-trivial results are only vaguely discussed, or their analysis is arbitrary. Therefore I suggest including a more detailed discussion of the results, that would address the following points:**

**First of all, we would like to thank the reviewer for the detailed and constructive feedback. With the help of this review, we hope to have considered all missing points and open questions in the revised version.**

1. Single cloud simulations of cumulus show, that splitting increases the amount of rain water. This increase is not seen in the cloud field simulation. Authors conclude that there is no increase in rain water because of averaging over a large cloud field (p.15 l.4). I do not understand how this is relevant. If amount of rain water in each cloud is increased, the average should also be increased.

**Author's answer:** This objection is correct. As you suggested in the next comment we ran for each of the reference simulations a small ensemble (for each case 5 ensembles), showing that differences in the RWP and Z can be lead back to different model realizations. In the revised manuscript in Fig. 15 (in former revision Fig. 13) for each case one model realizations is shown as well the range of the different model realizations.

**Modification** (**Ensembles of single cloud):** ~~All splitting configurations show higher RWPs in comparison to the reference runs without splitting.This increase of up to 12% is a direct result of the improved collisional growth process in the splitting configurations, resulting in more numerous and larger rain drops.This is also observed for the radar reflectivity (Fig. 11d), which is proportional to the second moment of the DSD and hence more sensitive to larger droplets.~~

In the reference simulations (represented as a mean of 5 ensemble for each case) of Figs. 15c and 15d, one can seen an increase in the precipitation parameters (RWP, radar reflectivity and precipitation sum) for an increased number of superdroplets. However, the differences among the ensembles members are quite large, which is shown by the range (gray area) and the band of plus-minus one standard deviation

from the mean (light blue area) derived from all 15 ensemble members. Overall, the splitting simulations have a slight tendency to compare better with the reference cases using 87 and 186 superdroplets. Admittedly, since the results are (for the most part) within one standard deviation, it can be concluded that splitting has no significant influence on the global precipitation parameters.
(**page 14 line 14**)

[...]

Considering the temporal variability of the precipitation rate and total precipitation (Fig. 17e and f), no significant changes are detectable using splitting or a very high number of superdroplets  (**page 16 line1**)

[...]

In the idealized single cloud simulation, splitting improved the representation of collisional growth with up to 70 % larger maximum radii .
(**page 16 line 31**)

[...]

Figure 15: Ensemble results added.

Timeseries of different variables for the idealized single cloud simulation for different initial numbers of superdroplets and splitting configurations. In (a), the ratio of the actual and initialized number of superdroplets in the whole model domain is shown. The liquid water path (LWP) and rainwater path (RWP) are displayed in panels (b) and (c), respectively. In (d), the total radar reflectivity is shown. Panels (e) and (f) show the precipitation rate and total precipitation, respectively. The reference simulations (runs without splitting) are presented as a mean of five ensembles for each case. Moreover, the light blue areas show the mean plus-minus one standard deviation and the gray areas show the range derived from all 15 ensemble members.

2. Judging from single cloud simulations, merging increases radar reflectivity and amount of precipitation (see Fig. 13, especially for the S20 case). I suppose that this is not the case, but that the difference comes from different model realizations. This could be clarified if a small ensemble of single cloud simulations was ran for each case discussed in the section 4.2.

**Author's answer:** As mentioned above, we agree with this comment and have adapted our manuscript (see comment above).

**Modification:** The extensive changes and results of the ensemble runs are summarized in the response to the first comment to which reference is made here.

3. In single cloud simulations without splitting, amount of precipitation decreases as the number of computational droplets is increased (Fig. 13). Simulations with

splitting are in better agreement with $N_p$= 15 than with Np= 186 .Why is it so?

**Author's answer:** This expression occurred due to using only one model realization. Different realizations show different results which are now considered in Fig. 15 (old Fig. 13).

**Modification:** The extensive changes and results of the ensemble runs are summarized in the response to the first comment above.

4. In box model simulations without splitting, largest droplets produced in LCM are larger than the Smoluchowski equation predicts, especially if number of computational particles is large (Fig. 5). On page 9, line 26, Authors argue that this is because there are few large SDs with high weighting factors. If this is the case, then shouldn't the effect decrease with increasing number of SDs? The opposite is observed - this effect is more pronounced as more SDs are added. Moreover, if the Authors' argument was correct, splitting of SDs should also fix this problem. On the contrary, LCM with splitting also produces too large droplets, as seen in Figs. 7 and 10.

**Author's answer:** The underlying problem here is the initialization, which cannot be fixed by splitting. The initialization with constant weighting factors will always deviate from the exponential initialization used by Wang et al. (2007). Therefore, subsequent collisions, which are improved by splitting, cannot agree with the bin-solution by Wang et al. (2007) whatsoever. Thus, the decreasing difference among the different LCM simulations, as it is occurring due to splitting, needs to be seen as a proof of concept, and not the comparison with the bin results.

**Modification** (**page 11 line 2):** Again, general differences between the models are caused by the (problematic) initialization, which cannot be fixed by splitting. The initialization with constant weighting factors will always deviate from the exponential initialization used by Wang et al. (2007). Therefore, subsequent collisions, which are improved by splitting, cannot agree with the bin-solution by Wang et al. (2007) whatsoever. Thus, the decreasing difference among the different LCM simulations, as it is occurring due to splitting, needs to be seen as a proof of concept, and not the comparison with the bin results.

5. In the first paragraph of "Conclusions", Authors claim that LCMs are "known to insufficiently represent" coalescence. I suppose that they are referring to results of box model simulations, shown in Fig. 1. Not all LCM insufficiently represent coalescence, but only those that initialize SDs in the same way it is done in the paper. Other initialization procedures give much better box model results, e.g. Dziekan and Pawlowska 2017.

**Author's answer:** Sorry for this misunderstanding. This statement is based on the insufficient representation of collision/coalescence in LCMs with a constant weighting factor, which is a typical way to initialize those models in three-dimensional applications. We definitely agree that LCMs with different initialization methods are

able to represent collision very well (e.g. in box models) even without splitting. However, in three-dimensional applications, it is not always possible to use such initialization methods (like singleSIP or multiSIP).

**Modification (page 16 line 12):** These models are able to represent collision and coalescence well (Unterstrasser et al., 2017;Dziekan and Pawlowska, 2017). Under certain conditions, however, they are known to insufficiently represented this process.

These conditions occur when the number of superdroplets is low and, accordingly, the number of real droplets represented by each superdroplet (the so-called weighting factor) is high, leading to an oversimplified representation of the droplet size distribution (DSD) (Riechelmann et al., 2012; Unterstrasser et al., 2017).
[...]
examples for LCM applications using a constant weighting factor/multiplicity (e.g. Shima et al., 2009; Riechelmann et al., 2012; Arabas and Shima, 2013; Naumann and Seifert, 2015, Hoffmann et al., 2017; Sardina et al., 2018) **(page 7 line 15)**

6. In "Conclusions", cloud field simulations are claimed to show similar effects of splitting on the production of rain as single cloud simulations. This is not true - splitting increases the amount of rain in single cloud simulations, but does not affect the amount of rain in cloud field simulations.

**Author's answer:** The ensembles show that our conclusion to macrophysical properties of a single cloud was not correct. We corrected that in the revised revision. Now it is in accordance that splitting does not change cloud-wide properties as rain water path or radar reflectivity, but has an influence on the spatial and temporal representation of rain droplets and the representation of the DSD.

**Modification:** The extensive changes and results of the ensemble runs are summarized in the response to the first comment to which reference is made here.

7. A short discussion of potential impact of SD merging on aerosol processing would be desirable.

**Author's answer:** Merging needs and can be adopted for future applications in which aerosol size and composition are considered. However, these more sophisticated approaches are not in the scope of the presented paper which focuses on the production of rain.

**Modification (SD merging on aerosol processing):** Furthermore, it must be mentioned that the merging algorithm described here, do not conserve size and chemical composition of the aerosol. Therefore, studies that explicitly simulate the activation process may have to adapt the merging algorithm. **(page 6 line 25)**

8. Authors write that merging of SDs reduces computing time by 18% . How does splitting affect performance?

**Author's answer:** A short answer is: the computing time increases about 0-20% depending on how many particles are created (for the idealized cloud setup). We observed that the simulation S20 requires 19.2% more computing time than the reference simulation const. $N_p87$. By applying splitting, the simulation S20 merging had nearly the same computational time than the reference simulation and was only 1.2% slower.

The storage demand can be estimated from Fig. 13a. The ratio of the actual number of superdroplets to the initialized number of superdroplets is a measure of the increased demand, where the highest increase can be observed at S10 which is about 15%.

**Modification (page 15 line 3):** To estimate the increase in computing time due to splitting, we conducted three simulations (const. $N_P87$, S20 and S20merging) with comparable time measurements. The constraint to three simulations is caused by a special mode which is required for time measurements on the supercomputer but leads to an increase in computing time. Here, we observe that a splitting-simulation S20 require 19.2% more computing time than the reference simulation const. $N_P87$. If applied, merging allows a massive reduction of the number of superdroplets, reducing the computing time by 18% and the storage demand (which is proportional to the number of superdroplets) by at least by 7% compared to simulations applying only splitting (Fig.13a). All in all, the simulation applying both splitting and merging, is only 1.2% slower than the reference simulation const. $N_P$ 87.

**Technical comments:**
9. In lower panels of Fig. 15, precipitation rate from vanZanten et al. 2011 should be shown.

**Author's answer:** The precipitation rate is added in the revised version.

**Modification (Figure 15):** The precipitation rate from vanZanten et al. 2011 is added.

10. Through the paper, large variability in results is alternatively called "oscillations" or "fluctuations" (e.g. p. 8, l. 19). Oscillation is a periodic process, so I suggest the word "fluctuations" to be used.

**Author's answer:** This is corrected in the revised version.

**Modification (page 9 line 11 and page 9 line 27):** "oscillations" → "fluctuations"

11. In the last line on page 3, Authors write that splitting makes their approach "fundamentally different". To me this seems to be a too strong statement - "different" would suffice.

**Author's answer:** This is corrected in the revised version.

**Modification (page 4 line 3):** [..] different[..]

12. Why is the blue line in Fig. 16 discontinuous?

**Author's answer:** To calculate the PDF of the precipitation rate, the rates were discretized. With a low number of superdroplets, some bins are empty due to a lack of statistics, which explains the discontinuity of the function.

13. In Fig. 5 spectra for 512 and 1000 SDs have a local maximum on the large end. What is the cause?

**Author's answer:** A closer analysis reveals this also the case with 15 and 37 superdroplets. The reason for this is the insufficient statistics (even with 500 and 1000 superdroplets per grid box) in the range of very large drops. The data shows that the last bins are represented by only 1-3 superdroplets (resulting from all 25.344 boxes).

---

## Author Comment (AC3) · 10 Aug 2018

**1 Introduction**

**The authors introduce a new extension of the lagrangian microphysics schemes- the splitting and merging algorithm. The splitting part of the algorithm increases the resolution of the lagrangian microphysics scheme for big droplets. This is especially important for correctly representing the collisions between droplets and the resulting onset of precipitation. The merging part of the algorithm decreases slightly the computational cost of the splitting algorithm. Both developments are described and tested. The paper is well written and interesting for the GMD community. It should be published after some corrections and additional tests.**

**First of all, we would like to thank the reviewer for the detailed and constructive feedback. With the help of this review we hope to have considered all missing points and open questions.**

**2 Major comments**

• A lagrangian microphysics scheme can represent collisions in different ways and can be initialized in different ways. Both choices have big impact on the accuracy of the scheme, Unterstrasser et al. (2017). Here the authors in their implementation of the lagrangian microphysics scheme use the most accurate way to represent collisions (the all-or-nothing algorithm Shima et al. (2009)). This is great. However, they initialize the droplet size distribution with constant weighting factors. This is the worst initialization strategy for representing collisions, see Figure 1 in Unterstrasser et al. (2017). It cannot be said that this is the standard way to initialize lagrangian microphysics schemes. Other groups initialize their schemes in different ways, see for example Unterstrasser and Sölch (2014), Arabas et al.(2015). Because this work focuses on improving the representation of collisions between droplets, such a bad-for-collisions initialization choice is not justifiable. The authors should also test their splitting and merging algorithm with a better initialization way (any of the singleSIP, multiSIP or v random from Unterstrasser et al.(2017) would suffice). Redoing the single cloud and cloud field simulations might be to expensive and unnecessary for the purpose of testing. However, it would be very interesting to see the single and multi-box tests done again with a different initialization choice. Does the splitting and merging algorithm improve the results as much for a different initialization choice Are the multi-box simulations really necessary if the droplet size distribution is initialized correctly? How does the estimate given in line 25 on page 15 change with a better initialization?

**Author's answer:** That objection is correct. We already performed single-box simulations with singleSIP initialization to validate our collision algorithm. These simulations show that we achieve very good results with 87 particles per grid box compared to the bin model. For this initialization, splitting only results in a slight

reduction of fluctuations compared to simulations with a small number of superdroplets. However, as described in the paper, in real 3D applications, as with an idealized box model, it is not guaranteed that large drops (relevant as collision embryos) are statistically sufficiently represented. In fact, many 3D LCM applications use the same weighting factor for all superdroplets in their simulations, which we tried to mimic in our box simulations.

**Modification (concerning different initialization):** As reference, the singleSIP initialization of Unterstrasser et al. (2017) is used for the single-box model, too. In contrast to the previously described initialization, the initial DSD is discretized using logarithmically spaced bins. The number of bins corresponds to the number of superdroplets. To each bin, a superdroplet with a corresponding mean radius and weighting factor is assigned. The maximum radius of the initial distribution is approximately 33 µm, which corresponds to a number of concentrations of $1/\Delta V$. This avoids superdroplets with a weighting factor less than 1. Note that this (not always applicable) initialization technique represents the inherent variability of droplet radii and their abundance across the initial spectrum much more accurately than the previously described method, and, therefore results in a much better agreement with literature
references. **(page 8 line 6)**
[...]
Figures 1 and 2 show the mass density distribution after 3600 s and the temporal development of the moments for the LCM applied as a single-box model using the singleSIP initialization by Unterstrasser et al. (2017). Each grid box is initialized with a different number of superdroplets (colored lines). The reference solution of Wang et al. (2007) is shown as a black solid line. Figure 1 shows that even with 87 superdroplets the solution of Wang et al. (2007) can be reproduced well and a further increase
in the number of superdroplets only leads to minor improvements. The small deviations between the bin model solution and the LCM can be traced back to the different solution of the collection equation (e.g., Dziekan and Pawlowska, 2017). Overall, it can be seen that the solution of the LCM converges with an increasing number of superdroplets. The moments of mass distribution (Fig. 2) also show convergence with an increasing number of superdroplets. This good representation of collision growth is in line with the results with Unterstrasser et al. (2017).
Now, Figs. 3 and 4 show the same quantities but for the initialization with identical weighting factors.
**(page 8 line 29)**

• **page 6 line 3:** What is more important in counterbalancing the increase in super-droplet number due splitting algorithm? - Is it the limiting number of super-droplet

per grid-box $N_{P,max}$ or merging algorithm? What is the impact of merging on the possible future activation of merged super-droplets to cloud droplets? What is the resulting resolution of the scheme for aerosol particles after merging? Is merging more accurate than using super-droplets just for representing clouds and precipitation and parameterizing activation process, as it is done in Grabowski et al. (2018)?

**Author's answer:** Both, the merging algorithm and the limitation of the number of superdroplets, are very important mechanisms to keep the number of superdroplets at a computationally feasible level. However, they work in different places. $N_{P,max}$ limits the number of superdroplets of a grid box inside the cloud, which is necessary to compute collisions in a reasonable amount of time (the performance of the collision algorithm is O(N^2)). Merging, on the other hand, nudges the superdroplet concentration outside the cloud to the initial superdroplet concentration. Overall, $N_{P,max}$ is a necessary constraint to be able to perform the simulation with splitting, while merging is a good supplement to save computing time.

In this study, the explicit representation of activation (which is a strength of LCMs) was neglected. At present, the merging algorithm would not take into account the size and chemical composition of aerosols. Although certain additions to the merging algorithm to cope with the size and chemical composition of aerosols are imaginable, they are out of the scope of the is manuscript.

In contrast to the approach of Grabowski et al. (2018), our approach would allow to track detailed changes in the number (and, in possible future applications, the size and composition) of aerosols. In that sense, merging is more accurate but needs more computing time and memory since the baseline number of superdroplets is higher.

**Modification (page 6 line 19):** See modifications to next comment and comment concerning "page 15 line 27".

• **page 6 line 16-19:** The mass conservation is a necessary constraint. But it is not enough to determine the super-droplet properties after merging. For the purpose of this study it's probably not important to have a more detailed strategy for determining super-droplet properties after merging. However it might be important for aerosol processing or secondary activation of aerosol particles to cloud droplets. Having that in mind, what would be the best way to determine the super-droplet properties after merging? For example: The current one assumes that the super-droplet radius after merging is equal to the radius of the super-droplet with the bigger weighting factor. An alternative could be to assume that the weighting factors are summed and to calculate the new radius from the mass conservation?

**Author's answer:** Thanks for that suggestion, we also did this. Furthermore, we merged the two most similar superdroplets, summing both weighting factors, and as you suggested, calculate the new radius. However, the results shows no difference to the current method except that the actually applied algorithm is computationally more efficient, which is a strong cause for the usage of the proposed method.

**Modification (page 6 line 19):** Moreover, a more advanced method where the most similar droplets (within one grid box) concerning their mass are merged was tested. Using this, simulations shows that there are no indications for different results using the advanced method. However, due to a sorting process the computing time is increased in comparison to the simple method.

[…]

Furthermore, it must be mentioned that the merging algorithm described here, do not conserve size and chemical composition of the aerosol. Therefore, studies that explicitly simulate the activation process may have to adapt the merging algorithm. **(page 6 line 25)**

• **page 7 line 11:** Are all the boxes in the multi-box simulation homogeneous and there is no droplet sedimentation? If yes, what is the difference between the introduced here multi-box approach and a single-box simulation that would use a better way to initialize the droplet size distribution (with a better initial representation of the tail of the distribution) and use the same total number of super-droplets as the multi-box simulation? What is the total volume simulated in the multi-box approach vs the single-box? Is multi-box approach increasing the simulated volume only to introduce different realizations of the initial condition to counter the problems introduced by the constant weighting factor initialization?

**Author's answer:** Yes, in the multi-box approach droplet sedimentation is not considered. The multi-box approach is an attempt to represent collisional growth with an unfortunate initialization (limited number of superdroplets per grid box and (high) constant weighting factor) under idealized conditions (box model simulation, with initial size spectrum), while allowing superdroplets to move between grid boxes as in a three-dimensional simulation. Of course you are right an unfettered comparison between the single-box and multi-box approach is not fair.

And we agree that the multi-box approach increases the simulated volume by introducing different realizations of the initial conditions, which partly counters the problems introduced by the initialization with a constant weighting factor. However, the relative differences are interesting. Seeing that in the multi-box approach splitting of the largest droplets is a powerful tool to mitigate a bad initialization and to represent the size droplet distribution is as good as it is with 1000 superdroplets (in the multi-box approach) initially.

When a different initialization is used (singleSIP from Unterstrasser et al (2017)), which is now displayed in Figs. 1 and 2, no splitting or multi-box approach is necessary to reach a high agreement with the bin reference. However, as shown in comment to page 7 line 6, initializing LCMs with a constant weighting factor is common in many LES-LCM applications.

**Modification (page 7 line 11):** Modifications to this comment are included in the changes related to the first comment of the reviewer (see first major comment).

• **Both the high super-droplet concentration simulations** (Fig 5) and the splitting simulations (Fig 7) overestimate the biggest droplet sizes. Why? Would using even more super-droplets allow to reach better agreement with the bin reference simulation for the biggest droplet sizes?

**Author's answer:** The underlying problem here is the initialization, which cannot be fixed by splitting. The initialization with constant weighting factors will always deviate from the exponential initialization used by Wang et al. (2007). Therefore, subsequent collisions, which are improved by splitting, cannot agree with the bin-solution by Wang et al. (2007) whatsoever. Thus, the decreasing difference among the different LCM simulations, as it is occurring due to splitting, needs to be seen as a proof of concept, and not the comparison with the bin results.

**Modification** (**page 11 line 2):** Again, general differences between the models are caused by the (problematic) initialization, which cannot be fixed by splitting. The initialization with constant weighting factors will always deviate from the exponential initialization used by Wang et al. (2007). Therefore, subsequent collisions, which are improved by splitting, cannot agree with the bin-solution by Wang et al. (2007) whatsoever. Thus, the decreasing difference among the different LCM simulations, as it is occurring due to splitting, needs to be seen as a proof of concept, and not the comparison with the bin results.

• **page 11 line 14:** What is the benefit of using a 3-dimensional simulation setup if the initial condition is 2-dimensional? Using a 2-dimensional setup (with cyclic boundary condition in the missing dimension) would reduce the computational cost by two orders of magnitude. This in turn would allow to test the performance, accuracy and convergence of the scheme for two orders of magnitude higher super-droplet concentrations.

**Author's answer:** This setup was used because it is well known and extensively tested by our group (Riechelmann et al., 2012, Hoffmann et al., 2013, Hoffmann et al., 2017). We agree that a full three-dimensional cloud setup is more appropriate and we will switch to such a setup in the future. However, using a pure two-dimensional setup would lead to a different evolution of turbulence, even though this may be negligible in our study. Moreover, since the computing time increases quadratic with the number of superdroplets per grid box and the parallelization is done in vertical columns the estimation that a two-dimensional setup would allow by

two orders of magnitude higher superdroplet concentrations is unfortunately not applicable to our model.

• **Figure 11:** I agree that eliminating the fluctuations in the size distribution of big droplets is a good result. However, the box model tests (Fig 5 and 7) show that the lagrangian scheme overestimates the sizes in the large tail of the droplet distribution. Therefore the fact that the biggest drop size for the simulation with splitting is 350µm bigger might not necessarily be an improvement?
**Author's answer:** Differences in the box-model simulation to the bin model are caused by the initialization (see comment to Fig. 5 and 7), which are specific to the box-simulations and the comparison with the bin-solution. Accordingly, the fact that the maximum droplet size is increased in more-dimensional simulations, should be seen as an improvement gained from the better representation of collision and coalescence.

• **Figure 11 and 12:** Is there any change in the behavior for the small droplet sizes at t=3000s (end of the simulation) that is caused by merging?
**Author's answer:** No, merging affects only superdroplets with radii smaller than 0.1 µm in non-cloudy regions. Accordingly, these particles contain too little water to affect any of the displayed quantities.

• **Figure 13 c, d, f:** Why is the simulation with the biggest initial concentration of super-droplets (N=186) the biggest outlier? Out of the simulations without split-ting and merging I would expect the one with initial N=15 to perform the worst and not the best. If the resolution for big droplets is important for collisions then the N=186 simulation should be better than N=15? Would running an ensemble average help? Or is this behavior consistent for even higher initial super-droplet concentrations?
**Author's answer:** This objection is correct. We decided to run an ensemble for each reference simulations (5 simulations for each case). They show that differences in the RWP and Z can be traced back to different model realizations.  In the revised manuscript (Fig. 15), we now show the ensemble mean and range of the different model realizations.
**Modification** (**Ensembles of single cloud):** ~~All splitting configurations show higher RWPs in comparison to the reference runs without splitting.This increase of up to 12% is a direct result of the improved collisional growth process in the splitting configurations, resulting in more numerous and larger rain drops.This is also observed for the radar reflectivity (Fig. 11d), which is proportional to the second moment of the DSD and hence more sensitive to larger droplets.~~

In the reference simulations (represented as a mean of 5 ensemble for each case) of Figs. 15c and 15d, one can seen an increase in the precipitation parameters (RWP,

radar reflectivity and precipitation sum) for an increased number of superdroplets. However, the differences among the ensembles members are quite large, which is shown by the range (gray area) and the band of plus-minus one standard deviation from the mean (light blue area) derived from all 15 ensemble members. Overall, the splitting simulations have a slight tendency to compare better with the reference cases using 87 and 186 superdroplets. Admittedly, since the results are (for the most part) within one standard deviation, it can be concluded that splitting has no significant influence on the global precipitation parameters.
(**page 14 line 14**)

[...]

Considering the temporal variability of the precipitation rate and total precipitation (Fig. 17e and f), no significant changes are detectable using splitting or a very high number of superdroplets   (**page 16 line1**)

[...]

In the idealized single cloud simulation, splitting improved the representation of collisional growth with up to 70 % larger maximum radii .
(**page 16 line 31**)

[...]

Figure 15: Ensemble results added.
 Timeseries of different variables for the idealized single cloud simulation for different initial numbers of superdroplets and splitting configurations. In (a), the ratio of the actual and initialized number of superdroplets in the whole model domain is shown. The liquid water path (LWP) and rainwater path (RWP) are displayed in panels (b) and (c), respectively. In (d), the total radar reflectivity is shown. Panels (e) and (f) show the precipitation rate and total precipitation, respectively. The reference simulations (runs without splitting) are presented as a mean of five ensembles for each case. Moreover, the light blue areas show the mean plus-minus one standard deviation and the gray areas show the range derived from all 15 ensemble members.

• **page 14 line 6:** What is the computational cost and storage demand increase due to splitting algorithm?
**Author's answer:** This question must be answered separately. A short answer is: the computing time increases about 0-20% in dependence of how many particles are created (for the idealized cloud setup). Here, we see that the simulation *S20* requires 19.2% more computational time than the reference simulation *const. $N_p$87*. By applying splitting the simulation *S20 merging* needed nearly the same computational time than the reference simulation and was only 1.2% slower.
The storage demand can be estimated from figure 13a. The ratio of the actual number of superdroplets to the the initialized number of superdroplets is a measure

of the increased demand, where the highest increase can be observed for *S10* which is about 15%.

**Modification (page 15 line 3):** To estimate the increase in computing time due to splitting, we conducted three simulations (const. $N_P87$, S20 and S20merging) with comparable time measurements. The constraint to three simulations is caused by a special mode which is required for time measurements on the supercomputer but leads to an increase in computing time. Here, we observe that a splitting-simulation S20 require 19.2% more computing time than the reference simulation const. $N_P87$. If applied, merging allows a massive reduction of the number of superdroplets, reducing the computing time by 18% and the storage demand (which is proportional to the number of superdroplets) by at least by 7% compared to simulations applying only splitting (Fig.13a). All in all, the simulation applying both splitting and merging, is only 1.2% slower than the reference simulation const. $N_P$ 87.

• **page 15 line 13:** The work by Dziekan and Pawlowska (2017) shows that when used in high-enough resolution the lagrangian methods truly do resolve the collisions between droplets. The tests presented in Unterstrasser et al. (2017) also suggest that when initialized correctly and when using a good algorithm for representing collisions the lagrangian microphysics schemes can represent collisions for coarser resolution settings (i.e low initial super-droplet concentration). The splitting and merging algorithm presented here is a valid improvement. It is especially important for large eddy simulation applications when by necessity the lagrangian schemes have to be used with low super-droplet concentrations. Nevertheless in my opinion, saying that in general the lagrangian methods are known to insufficiently represent collisions is not justified.

**Author's answer:** Sorry for this misunderstanding. This statement is based on the insufficient representation of the collision growth of LCMs under certain initializations when a constant and large weighting factor is used. We definitely agree that LCMs with different initialization methods are able to represent collision very well (e.g. in box models) even without splitting. However, in certain applications it is not always possible to use such initialization methods (like singleSIP or multiSIP).

**Modification (page 16 line 12):** These models are able to represent collision and coalescence well (Unterstrasser et al., 2017;Dziekan and Pawlowska, 2017). Under certain conditions, however, they are known to insufficiently represented this process.

These conditions occur when the number of superdroplets is low and, accordingly, the number of real droplets represented by each superdroplet (the so-called weighting factor) is high, leading to an oversimplified representation of the droplet size distribution (DSD) (Riechelmann et al., 2012; Unterstrasser et al., 2017).

**3 Minor comments**

• The Lagrangian particles used to represent droplets are named differently by different modeling groups: super-droplets, simulation particles, etc. The original term super-droplet was introduced by Shima et al. (2009). Instead of using another notation superdroplet it would be better to follow the notation that is already used by others.

**Author's answer:** This was our attempt to establish the word, after consultation with native English speakers, in its the most correct variant.

• **page 1 line 16:** Are the references meant to be chronologically or alphabetically ordered?

**Author's answer:** In the revised version they are chronologically ordered.

**Modification (page 1 line 16):** (Shima et al., 2009; Sölch and Kärcher, 2010; Andrejczuk et al., 2010; Riechelmann et al., 2012; Arabas et al., 2015; Naumann and Seifert, 2015; Grabowski et al., 2018; Sardina et al., 2018)

• **page 1 line 16:** A couple more references to lagrangian microphysics applications: Lee et al. (2014), Arabas et al. (2015), Sardina et al. (2018)

**Author's answer:** Arabas et al. (2015) and Sardina et al. (2018) added in revised version. In our mind the citation of Lee et al. is not a useful citation.

**Modification (page 1 line 16):** (Shima et al., 2009; Sölch and Kärcher, 2010; Andrejczuk et al., 2010; Riechelmann et al., 2012; Arabas et al., 2015; Naumann and Seifert, 2015; Grabowski et al., 2018; Sardina et al., 2018)

• **page 1 line 23**: Which of the previously cited works use all-or-nothing algorithm?

**Author's answer:** The basic idea of the all-or-nothing algorithm is based on Shima et al. (2009) and Sölch and Kärcher (2010). Moreover this approach is used in the works of Arabas et al. (2015), Dziekan and Pawlowska (2017), and Hoffmann et al. (2017).

**Modification (page 1 line 24):** (based on Shima et al. (2009) and Sölch and Kärcher (2010), and used by Arabas et al. (2015), Dziekan and Pawlowska (2017), and Hoffmann et al. (2017)) exhibits the best performance, i.e., it agrees well with analytical solutions or other modeling approaches used to represent collection. Using an unfortunate initialization of superdroplets with equal weighting factors, however, even the all-or-nothing algorithm struggles to represent the precipitation process correctly

• **page 2 line 3**: What is a large weighting factor and a large number of super-droplets? Could you provide an order of magnitude estimate of those for a typical large eddy simulation grid box?

**Author's answer:** A large weighting factor is in the magnitude of $10^9$. Typical LES applications with a grid size in the order of 10m and a typical super-droplet

concentration of 100 per grid box need such weighting factors to represent number concentrations of 100 cm$^{-3}$. Even though the weighting factor changes due to collision and coalescence the reduction is insufficient for large droplets (r>100µm) to represent rain droplets statistically appropriate. Consequently, it is essential to have a large amount (in the magnitude of 10-100 per grid-box) of superdroplets representing large droplets with small weighting factors.

**Modification (page 2 line 6):** [..](with accordingly large weighting factors, approximately 10$^9$ for typical LES-LCM applications)[..].

[..](in the magnitude of 10-100 per grid-box with accordingly small weighting factors)[..].

• **page 2 line 30:** Is it more correct to say that it is a probability that super droplet m will collect super droplet n?

**Author's answer:** No, since the number of superdroplets does not change due to collisions and coalescence.

• **page 2 line 30:** An alternative is to consider for collisions only the non-overlapping pairs and scale the probability - see section 5.1.3 in Shima et al. (2009) or section 5.1.4 in Arabas et al. (2015). This allows for the collision algorithm to scale linearly and not quadratically with the number of super-droplets. Maybe it should be mentioned?

**Author's answer:** In our opinion, this is a too detailed discussion and out of focus of the manuscript.

• **page 6 line 14:** Are the super-droplets sorted with regard to their size in memory? Or in other words are the super-droplets merged with the most similar super-droplet in a given grid-box?

**Author's answer:** No, super-droplets are sorted with regard to their position in a given grid-box. This is due to an optimized method for the interpolation of velocities to particle positions (see Marong et al. 2015). Nevertheless, since only particles with radii smaller than 0.1µm are merged in this study, the effects can can be neglected. If explicit activation is considered merging should be modified.

**Modification (page 6 line 19):**Moreover, a more advanced method was tested, in which the most similar droplets (within one grid box) concerning their mass are merged.

These simulations show no different results. However, due to sorting processes the computing time is increased in comparison to the simple method.

• **page 6 line 26:** Is diffusional growth allowed in the box model simulations? If yes, what is the assumed saturation? It is confusing with regard to line 15 on page 12. - Saying that collisions dominate the droplet growth in the box simulations suggests that there are other processes considered.

**Author's answer:** No, diffusional growth is not allowed in the box-model simulations. We think there must be a misunderstanding: Page 12 line 15 refers to the single cloud setup (not the single box setup), where we are simulating an idealized shallow cumulus cloud in form of a rising warm air bubble. In that case diffusional growth is considered as well as collisional growth and all (thermo-)dynamics.
**Modification (page 6 line 30):** Therefore, the box model simulation consider collection as the only microphysical process.

• **page 6 line 30 (and onward):** It's a bit confusing to talk about grid boxes when using a single box model setup. There is no real computational grid here.
**Author's answer:** We agree for the single-box method. However, for the multi-box approach grid properties cannot be neglected. Talking about grid boxes is valid for consistency reasons.
**Modification (page 7 line 2):** Although zero-dimensional simulations do not have a spatial extent, allocating a certain weighting factor requires a reference volume to represent a defined droplet concentration.

• **page 6 line 30:** Where does the assumption that the box dimensions are $\Delta x = \Delta y = \Delta z = 20$m matter for the lagrangian scheme? Is it enough to say that the box volume is $8*10^3 m3$?
**Author's answer:** We partially agree to this comment.
**Modification (page 7 line 3):** Therefore, the volume of a grid box is $8 \cdot 10^3$ m$^3$ , which corresponds to an isotropically spaced grid with $\Delta x = \Delta y = \Delta z = 20$ m.

• **page 7 line 1:** How many boxes are used? How are the boxes in multi-box approach located with regard to each other? What are the boundary conditions?
**Author's answer:** In the single-box approach, an ensemble of 25,344 grid-boxes is simulated. For the Multi-Box approach also 25,344 grid-boxes are used. However, they a horizontal exchange of super-droplets due to sub-grid scale velocities is allowed. Lateral boundary conditions are cyclic.
**Modification (page 7 line 8):** In contrast to the calculation of in- dependent grid boxes, the multi-box approach allows superdroplets to move from one grid box to the next by prescribing a stochastic velocity (but no mean motion) in (7), using 25,344 grid boxes, as in the single-box ensemble above, with cyclic boundary conditions among which the superdroplets are allowed to move.

• **page 7 line 6:** As stated in my first major comment - I don't agree that the initialization with constant weighting factors is a standard. Because it is such a bad initialization for representing collisions the new splitting algorithm should also be tested with a better initialization.
**Author's answer:** In the reviewed version we show some box-simulation results with a different initialization. However, the method of choice for "real cloud" applications is

to start with an a certain superdroplet concentration and a constant weighting factor since it is unknown which particles will grow the most. Initializing the superdroplets with randomly chosen weighting factors or with weighting factors considering a prescribed aerosol spectrum also do not solve the problem, that large droplets are represented with unrealistic high weighting factors.

Therefore, we think our work provides useful information for the community.
Some examples for using constant weighting-factor in LCM applications are added as citations.

**Modification (page 7 line 15):** (e.g. Shima et al., 2009; Riechelmann et al., 2012; Naumann and Seifert, 2015, Hoffmann et al., 2017; Sardina et al., 2018)

• **Figure 2b:** Why is the LCM1000 behaves like a step function after 2500s?
**Author's answer:** The second moment is very sensitive to large radii of super-droplets. Collisions are discontinuous events and can produce significantly large droplets from one time-step to the next one.

• **Figure 3:** The plotting colors and patterns should be kept the same between Figs. 3,and 7 to allow easier comparison.
**Author's answer:** We agree.
**Modification (Figure 5):** Fig. 5 is changed.

• **page 10 line 25:** Is the initial super-droplet concentration again 87 per box?
**Author's answer:** Yes, it is!
**Modification (page 11 line 20):** Furthermore, all simulation are initialized identical with $N_{init}$= 87 superdroplets per grid box.

• **Figure 13a and 15c:** The notation $N_{SIP}$ was never used before. The authors choose to refer to the lagrangian particles as super-droplets and not simulation particles.
**Author's answer:** Thanks, corrected in the reviewed revision!
**Modification (Figure 13a and 15c):** Changed axes in reviewed version to $N_p$.

• **page 14 line 15:** When it is not possible? What happens then?
**Author's answer:** Removed in revised version. This phrase was aimed at superdroplet specific quantities (such as initial number of super-droplets/to actual number of superdroplets), which were obviously not calculated in vanZanten et al (2011).
**Modification (page 15 line 15):** Moreover, the calculation of the domain-averaged quantities follows (if possible) the descriptions given in the original case.

• **page 16 line 24:** Dziekan and Pawlowska (2017) would be a valid reference here.
**Author's answer:** We agree and it is added in the reviewed revision.

**Modification (15 line 24):** (Shima et al., 2009; Riechelmann et al., 2012; Unterstrasser et al., 2017; Dziekan and Pawlowska, 2017)

• **page 15 line 27:** Figure 10 suggests that the maximum number of super-droplets $N_{P,max}$ is as important as the splitting radius $r_{spl}$?
**Author's answer:** That could be the case if the initialized number of super-droplets is not taken into account. Here, 87 particles per grid-box are used initially. Therefore, it can be concluded that it is important that the threshold $N_{P,max}$ is not to close to the initial value. However, this also indicates that, if $N_{init}$ is large enough, it does not change the results.
**Modification (page 11 line 24):** Since the initial superdroplet concentration is $N_{init}=$ 87 for all configurations of $N_{P,max}$, it can be concluded that $N_{P,max}$ is a necessary but not crucial parameter as long as $N_{P,max} \geq 150$.

• **page 16 line 13:** The code of the Large Eddy Simulation model along with the new splitting and merging algorithm is available online and therefore fulfills the GMD requirements. It would have been great if the simple box model tests were available as a stand alone and easy to download and compile project. It would enable easy testing of the algorithm by others, for example this reviewer. It is by far too much coding to ask to do this now. I would just like to leave this comment as an idea for future development and testing.
**Author's answer:** Thanks for this comment. For future work we will heed this and will offer also our analysis tools and test cases which were developed and provide them in an own branch.